# ASXL2 is essential for haematopoiesis and acts as a haploinsufficient tumour suppressor in leukemia

Jean-Baptiste Micol[1,2,3,*], Alessandro Pastore[3,*], Daichi Inoue[3,*], Nicolas Duployez[4], Eunhee Kim[5], Stanley Chun-Wei Lee[3], Benjamin H. Durham[3], Young Rock Chung[3], Hana Cho[3], Xiao Jing Zhang[3], Akihide Yoshimi[3], Andrei Krivtsov[6], Richard Koche[7], Eric Solary[1,2], Amit Sinha[8], Claude Preudhomme[4] & Omar Abdel-Wahab[3,9]

Additional sex combs-like (ASXL) proteins are mammalian homologues of additional sex combs (Asx), a regulator of trithorax and polycomb function in *Drosophila*. While there has been great interest in *ASXL1* due to its frequent mutation in leukemia, little is known about its paralog ASXL2, which is frequently mutated in acute myeloid leukemia patients bearing the RUNX1-RUNX1T1 (AML1-ETO) fusion. Here we report that ASXL2 is required for normal haematopoiesis with distinct, non-overlapping effects from ASXL1 and acts as a haploinsufficient tumour suppressor. While *Asxl2* was required for normal haematopoietic stem cell self-renewal, Asxl2 loss promoted AML1-ETO leukemogenesis. Moreover, ASXL2 target genes strongly overlapped with those of RUNX1 and AML1-ETO and ASXL2 loss was associated with increased chromatin accessibility at putative enhancers of key leukemogenic loci. These data reveal that *Asxl2* is a critical regulator of haematopoiesis and mediates transcriptional effects that promote leukemogenesis driven by AML1-ETO.

[1] Hematology Department, Inserm UMR1170, Gustave Roussy Cancer Campus Grand Paris, Villejuif 94800, France. [2] Université Paris-Sud, Faculté de Médecine, Le Kremlin-Bicêtre, Paris 94270, France. [3] Human Oncology and Pathogenesis Program, Memorial Sloan-Kettering Cancer Center and Weill Cornell Medical College, New York 10065, USA. [4] Laboratory of Hematology and Tumor Bank, INSERM UMR-S 1172, Cancer Research Institute of Lille, CHRU of Lille, University Lille Nord de France, Lille 59037, France. [5] School of Life Sciences, Ulsan National Institute of Science and Technology, Ulsan 44919, Republic of Korea. [6] Dana-Farber Cancer Institute, and Division of Hematology/Oncology, Boston Children's Hospital, Harvard Medical School, Boston, Massachusetts 02215, USA. [7] Cancer Biology and Genetics Program, Memorial Sloan Kettering Cancer Center, New York, New York 10065, USA. [8] Basepair, Inc., New York, New York 10011, USA. [9] Leukemia Service, Dept. of Medicine, Memorial Sloan Kettering Cancer Center, New York, New York 10065, USA. * These authors contributed equally to this work. Correspondence and requests for materials should be addressed to O.A.-W. (email: abdelwao@mskcc.org).

Regulation of gene expression through the function of polycomb and trithorax protein complexes is critical for normal haematopoiesis[1–5]. The discovery of frequent mutations in the polycomb-associated protein additional sex combs-like, *ASXL1* (refs 6–8), resulted in efforts to understand its role in haematopoiesis. However, *ASXL1* is one of three mammalian paralogues of Asx, a protein required for both activation and silencing of *Hox* genes in *Drosophila*. Asxl2, in particular, shares a common expression pattern in embryogenesis and haematopoiesis with Asxl1 (refs 9–11), while Asxl3, in contrast, is not expressed in haematopoietic cells[12]. Moreover, ASXL1 and ASXL2 are encoded from paralogous regions of the genome at the *DNMT3B-ASXL1-KIF3B* (20q11.21) and *DNMT3A-ASXL2-KIF3C* (2p23.3) loci, respectively. Given these features and the fact that ASXL1 and ASXL2 also share common domains, it has been suggested that ASXL1 and ASXL2 may have overlapping, partially redundant, or even opposing functions[13].

*ASXL2* has recently been identified as among the most commonly mutated genes in AML patients bearing the AML1-ETO fusion oncoprotein[14–16] (encoded by translocation t(8;21)). At the same time, unlike mutations in *ASXL1*, *ASXL2* mutations are virtually absent in other forms of leukemia or in clonal haematopoiesis[17–19], suggesting a specific genetic interaction between *ASXL2* and *AML1-ETO*. Given that expression of the AML1-ETO fusion is not sufficient to generate overt AML on its own[20–24], the unique enrichment of *ASXL2* mutations in this subset of AML suggests that *ASXL2* mutations may be an important cooperating genetic alteration in the pathogenesis of AML1-ETO AML. Currently however, very little is known about the role of ASXL2 in normal or malignant haematopoiesis. Here we set out to define the role of ASXL2 in normal and malignant haematopoiesis, to compare its effects on gene expression and chromatin state to those of ASXL1, and to understand the functional basis for *ASXL2* mutations in the context of *AML1-ETO*-mediated AML. We identify that *Asxl2* is required for normal haematopoietic stem cell self-renewal and has non-redundant roles with Asxl1. Moreover, loss of Asxl2 altered chromatin state at key leukemogenic loci in AML1-ETO-expressing cells and loss of even a single copy of Asxl2 promotes AML1-ETO leukemogenesis.

## Results

### *ASXL2* mutations result in loss of ASXL2 protein expression.
Four recent studies have identified that *ASXL2* mutations are present in 19.8–22.7% of pediatric and adult patients with t(8;21) AML[14–16,25]. Across these studies, mutations in *ASXL2* occured as out-of-frame insertion/deletion mutations and nonsense mutations in exons 11–12 of *ASXL2*. To evaluate the effects of *ASXL2* mutations on ASXL2 expression, we generated mammalian expression vectors encoding full-length wild-type (WT) ASXL2 (1–1435) and two ASXL2 mutations documented in AML[15] (ASXL2 p.T740NfsX16 and p.E1287X). Expression of WT ASXL2 cDNA or cDNA constructs bearing leukemia-associated mutant forms of ASXL2 revealed reduced stability of mutant ASXL2 relative to WT ASXL2, with greater loss of mutant ASXL2 following cycloheximide exposure (Supplementary Fig. 1a,b). These data suggest that mutations in ASXL2 are most likely loss-of-function mutations.

### Generation of *Asxl2* conditional knockout mice.
Prior efforts to study the effects of ASXL2 loss *in vivo* have been limited by the fact that constitutive deletion of Asxl2 is associated with substantial perinatal lethality[10] and, to date, no models allowing complete post-natal or conditional, tissue-specific deletion of *Asxl2* have been established. Given the above data, and the fact

that human ASXL2 shares 79.4% total amino acid homology with mouse Asxl2, we generated a conditional allele targeting *Asxl2* (Fig. 1a and Supplementary Fig. 1c). Embryonic stem cell targeting was used to insert two LoxP sites flanking exon 11 of *Asxl2*, as well as a Frt-flanked neomycin selection cassette in the upstream intron. The generated mice (*Asxl2fl/fl*) were initially crossed to a germline Flp-deleter mouse line to remove the neomycin cassette and then subsequently crossed to IFN-α-inducible *Mx1*-cre transgenic mice to allow for inducible deletion of Asxl2 in post-natal tissues. Western blotting (WB) and quantitative PCR with reverse transcription of bone marrow (BM) mononuclear cells (MNCs) from *Mx1*-cre *Asxl2fl/fl* mice 4 weeks after administration of polyinosinic:polycytidylic acid (pIpC) revealed that Asxl2 protein expression was fully eliminated in this model and that Asxl2 deletion was not associated with altered Asxl1 expression at the protein or mRNA level (Fig. 1b,c and Supplementary Table 1).

### Asxl2 is required for normal haematopoiesis.
We next sought to delineate the role of Asxl2 on haematopoiesis and its potential redundancy with Asxl1. We and others have previously noted that Asxl1 loss in post-natal haematopoiesis is associated with modest leukopenia, anaemia and a myelodysplastic syndrome-like phenotype in mice after a long latency[26,27]. We therefore generated CD45.2 *Mx1*-cre control, *Mx1*-cre *Asxl2fl/fl*, *Mx1*-cre *Asxl1fl/fl* and compound *Mx1*-cre *Asxl1fl/fl Asxl2fl/fl* mice. To study the cell-autonomous effect of Asxl1/2 deletion in a haematopoieitic-specific manner, we performed noncompetitive BM transplantation (BMT) using these mice ($n = 3$–5 donors per genotype) into 10 lethally irradiated CD45.1 recipient mice per donor (Fig. 1d). This was then followed by peripheral blood analysis of the recipient mice and pIpC administration to delete *Asxl2* and/or *Asxl1* in the haematopoietic system.

Analysis of peripheral blood of *Asxl2*-deficient mice revealed that in contrast to the modest effects of *Asxl1* deletion on white blood cell (WBC) and platelet counts, *Asxl2* deletion resulted in rapid leukopenia and thrombocytopenia in recipient mice which was associated with hastened death of recipient mice (Fig. 1e–g). Moreover, mice with compound deletion of *Asxl1* and *Asxl2* had even lower WBC and platelet counts than single knockout controls. The decrease in peripheral blood MNCs in mice with *Asxl2* and compound *Asxl1/2* deletion was mostly attributed to decrements in B220$^+$ mature B-cells and CD3$^+$ T-cells (Supplementary Fig. 1d,e). Moreover, the lower peripheral blood platelet numbers in *Asxl2*-deficient mice was associated with significant decrease in the number of BM megakaryocytes as well as reduced colony formation of megakaryocyte progenitors purified from haematopoietic stem and progenitor cells (HSPCs) from *Asxl2*-deficient mice relative to Asxl1-deficient or control Asxl1/2-WT mice (Supplementary Fig. 1f–g).

Morphological analysis of BM histology sections revealed abnormal BM clustering of megakaryocytes that is consistent with defective megakaryocyte differentiation in *Asxl2*-deficient mice (Supplementary Fig. 2). Moreover, *Asxl2*-deficient mice exhibited hyposegmented neutrophils with hypogranular cytoplasm and circulating, multinucleated erythroid progenitors, which are features consistent with myelodysplasia (Supplementary Fig. 2b,c). *Asxl1*-deficient BM showed morphologic evidence of dysplasia in myeloid cells and megakaryocytes as previously reported[26,27], a feature that was also evident in mice with compound *Asxl1/2* deletion (Supplementary Fig. 2a). Of note, longitudinal evaluation of recipient mice transplanted with *Mx1*-cre *Asxl2fl/fl* mice up to 52 weeks failed to reveal any Asxl2-deficient mice to have developed increased white blood counts, splenomegaly or expansion of HSPCs

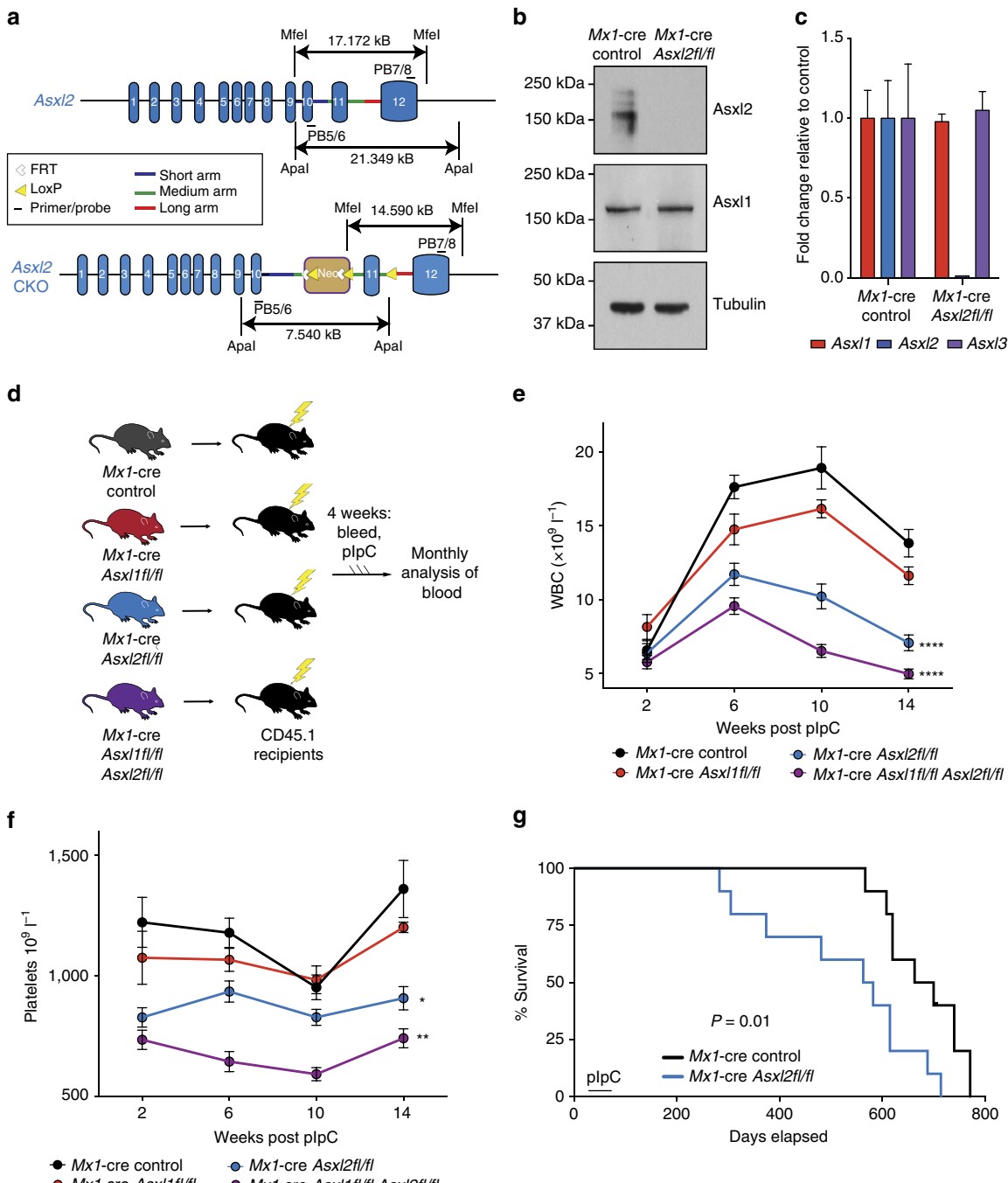

**Figure 1 | Asxl2 is required for normal haematopoiesis.** (**a**) *Asxl2* conditional knockout allele. (**b**) Western Blot of bone marrow (BM) cells from *Mx1*-cre control and *Mx1*-cre *Asxl2fl/fl* primary mice 4 weeks post-polyinosinic:polycytidylic acid (pIpC) adminstration. (**c**) Expression of *Asxl1*, *Asxl2*, and *Asxl3* by qRT-PCR in BM mononuclear cells from *Mx1*-cre *Asxl2fl/fl* primary mice 4 weeks post-pIpC relative to each gene from *Mx1*-cre controls. (**d**) Schema for noncompetitive BM transplantation using CD45.2[+] *Mx1*-cre control (black), *Mx1*-cre *Asxl2fl/fl* (red), *Mx1*-cre *Asxl2fl/fl* (blue), and *Mx1*-cre *Asxl2fl/fl Asxl2fl/fl* (purple) mice. (**e**–**f**) Enumeration of white blood cells (WBC) (**e**) and platelets (**f**) in peripheral blood of CD45.1 recipient mice following noncompetitive BM transplantation from CD45.2[+] mice as shown in (**d**) (n = 10 mice/genotype; pIpC was administered to recipient mice 4 weeks following transplantation). (**g**) Kaplan-Meier survival curve of recipient mice transplanted with *Mx1*-cre control and *Mx1*-cre *Asxl2fl/fl* BM cells (n=10 mice/genotype). Error bars represent mean ± s.d. *$P < 0.05$, **$P < 0.001$, ****$P < 0.0001$; *P*-values calculated by ordinary one-way ANOVA test.

(Supplementary Fig. 3). This finding suggests that Asxl2-deficient haematopoietic cells result in functional defects in haematopoiesis that are associated with shortened survival in the mice. Overall, these data reveal that Asxl2 is required for normal haematopoiesis and has distinct and non-redundant effects compared with Asxl1 in haematopoiesis.

**Asxl2 is required for haematopoietic stem cell self-renewal.** We next assessed the effect of *Asxl2* or compound *Asxl1/2* loss on HSPC frequency and function. We first performed serial competitive BMT assays using 500,000 BM MNCs from 6-week-old CD45.2 *Mx1*-cre control, *Mx1*-cre *Asxl2fl/fl*, *Mx1*-cre *Asxl1fl/fl* and compound *Mx1*-cre *Asxl1fl/fl Asxl2fl/fl* mice

pre-pIpC injection. These were injected with equal numbers of CD45.1 BM MNCs into lethally irradiated CD45.1 recipients (10 recipients per genotype) (Fig. 2a). Recipient mice were then bled 4 weeks later to establish baseline chimerism followed by

pIpC injection and monthly bleeding for 16 weeks post pIpC, following which mice were killed and serial transplantation was performed. Consistent with the effect of Asxl2 deletion on haematopoiesis in noncompetitive transplantation, Asxl2 loss was

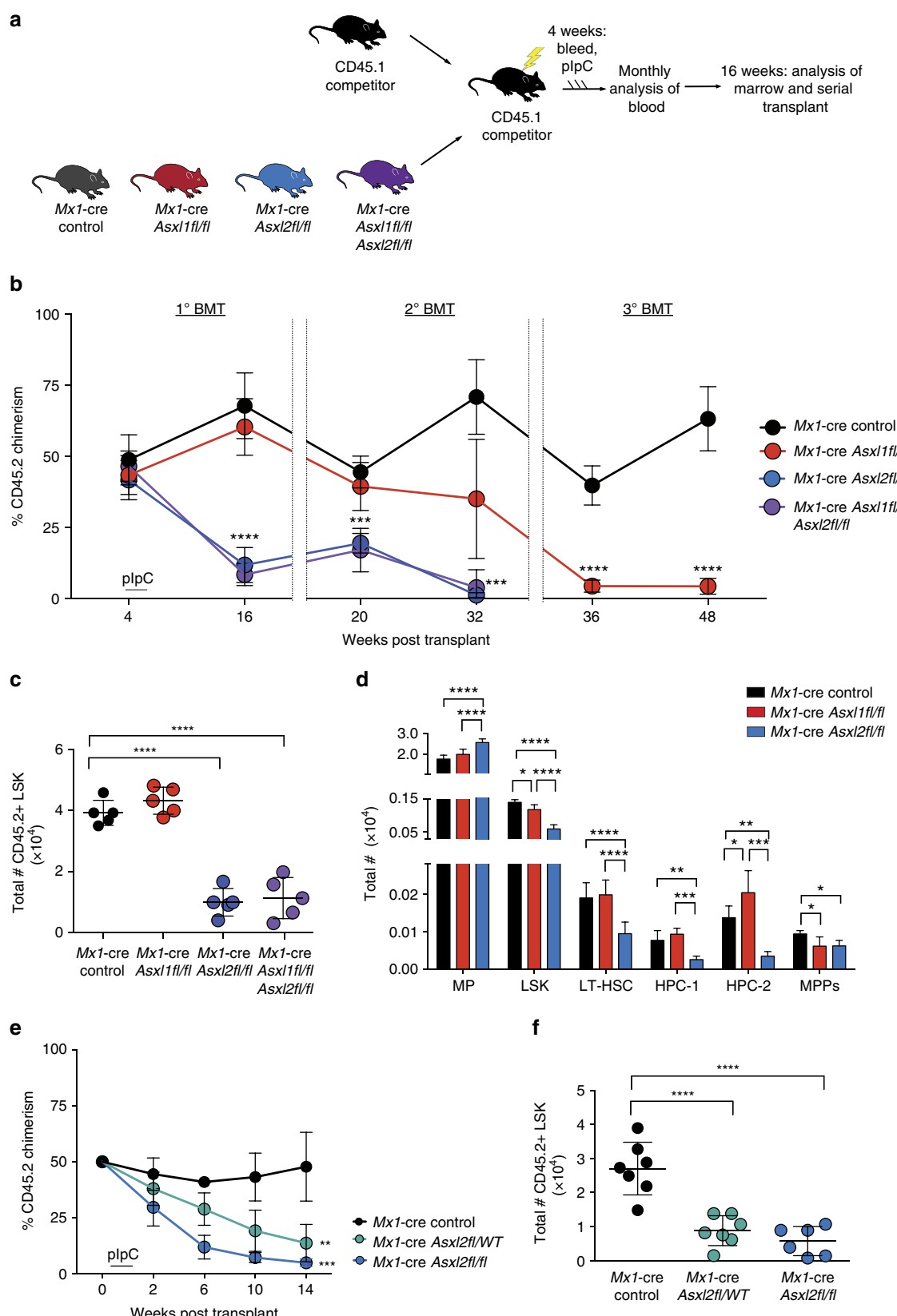

associated with rapid loss of haematopoiesis at both the level of mature circulating blood leukocytes and BM HSPCs (Fig. 2b,c). The effects of Asxl1 loss on HSPCs, in contrast, were evident only on serial transplantation. At 16-weeks post transplantation, Asxl2 loss was associated with loss of long-term haematopoietic stem cells (lineage-negative Sca−1+ c-Kit+ CD150+ CD48− cells) as well as more intermediate haematopoietic progenitor cell populations (haematopoietic progenitor fraction 1 and 2 (HPC−1 and HPC−2)[28]), phenotypes not seen with Asxl1 loss (Fig. 2d and Supplementary Fig. 4a).

Given that *ASXL2* mutations occur as heterozygous mutations in leukemia patients[14,15], we also investigated the effects of heterozygous loss of *Asxl2*. Competitive transplantation of *Mx1*-cre *Asxl2fl/WT* mice versus *Mx1*-cre *Asxl2fl/fl* and *Mx1*-cre control mice revealed that deletion of even a single copy of *Asxl2* was associated with substantial decrements in HSPC self-renewal (Fig. 2e,f). These data reveal that Asxl2 is haploinsufficient with regards to normal HSPC function.

Of note, the effects of *Asxl2* loss in the noncompetitive transplantation setting were also seen in the setting of primary, non-transplanted *Mx1*-cre *Asxl2fl/fl* mice relative to controls (Supplementary Fig. 4b–f). A cohort of 6 primary *Mx1*-cre *Asxl2* WT, *Mx1*-cre *Asxl2fl/WT* and *Mx1*-cre *Asxl2fl/fl* mice were treated with pIpC at 6 weeks of age and then followed for 26 weeks post pIpC. *Mx1*-cre *Asxl2fl/WT*, and *Mx1*-cre *Asxl2fl/fl* mice consistently exhibited lower periperhal blood WBCs (primarily due to lower B220+ cells in the peripheral blood) as well as platelet counts compared to *Mx1*-cre *Asxl2* WT controls (Supplementary Fig. 4b,c). Consistent with results in transplantation setting, significant reductions in absolute numbers of multipotent progenitors (MPPs), LSK cells and megakaryocyte/erythroid progenitor (MEP) cells as well as percentage of splenic erythroid precursors (CD71+ Ter119− cells) and megakaryocytes (CD41+ cells) were seen in *Mx1*-cre *Asxl2fl/fl* mice compared with *Mx1*-cre control mice (Supplementary Fig. 4d–f). *Mx1*-cre *Asxl2fl/wt* mice also exhibited significant reductions in absolute numbers of MEPs as well as splenic erythroid precursors (CD71+ Ter119− cells) and megakaryocytes (CD41+ cells) compared with *Mx1*-cre control mice (Supplementary Fig. 4d–f). These data further affirm that Asxl2 is required for normal haematopoiesis.

**Asxl2 target genes overlap with those of RUNX1 and AML1-ETO.** We next sought to understand the mechanistic effects of *Asxl2* loss on haematopoiesis by analysing the transcriptional and epigenomic effects of *Asxl2* loss. We performed mRNA sequencing (RNA-seq) of purified BM CD45.2+ lineage-negative Sca−1+ c-Kit+ (LSK) cells from *Mx1*-cre *Asxl2fl/fl*, *Mx1*-cre *Asxl1fl/fl,* and *Mx1*-cre control mice 4-weeks following pIpC administration. In contrast to the 129 genes significantly (multiple hypothesis adjusted $P < 0.05$) differentially expressed in *Asxl1*-deficient LSK cells relative to controls, 2,986 genes were dysregulated in *Asxl2*-deficient LSK cells relative to controls

(Fig. 3a and Supplementary Data 1). Moreover, only a small number of dysregulated genes were shared amongst *Asxl1*- versus *Asxl2*-null LSK cells (Fig. 3b). Interestingly, genes differentially expressed following Asxl2 loss significantly overlapped with genes previously described as upregulated by the AML1-ETO oncoprotein[29], a finding not seen amongst differentially expressed genes from *Asxl1*-deficient LSK cells (Fig. 3c). AML1-ETO is a transcriptional regulator whose genome-wide direct binding targets have been extensively mapped in human t(8;21) AML cell lines[30–33] and are known to overlap with binding targets of wild-type RUNX1 (refs 33,34). We therefore next examined the overlap between genes differentially expressed in *Asxl2*-deficient LSK cells and those which are known targets of AML1-ETO or RUNX1 based on previously published chromatin immunoprecipitation next-generation sequencing (ChIP-seq) studies in *ASXL1/2*-WT AML1-ETO-expressing human AML cells[29,30]. These data again revealed substantial overlap of direct AML1-ETO and RUNX1 gene targets amongst genes altered by *Asxl2* deletion, but not by *Asxl1* deletion (Fig. 3c).

**Overlap of ASXL2 and AML1-ETO target genes in AML cells.** The above data suggested an overlap of Asxl2-binding sites and transcriptional targets of AML1-ETO in the setting of normal haematopoiesis and without introduction of the AML1-ETO oncofusion protein. We therefore next sought to examine the transcriptional and epigenetic effects of ASXL2 loss in the setting of human AML expressing endogenous AML1-ETO. Comparison of ASXL2 genome-wide binding to RUNX1 and AML1-ETO-binding sites in SKNO−1 cells (a human AML1-ETO cell line WT for *ASXL1* and *ASXL2* (in contrast to the commonly studied Kasumi series of AML1-ETO-expressing cell lines which bear endogenous *ASXL1* mutation[35])) revealed that ASXL2 displayed less enrichment at transcriptional start sites (TSS) than RUNX1 or AML1-ETO (Fig. 3d). Given the likely different enrichment efficiencies between antibodies used in ChIP-seq here and the different number of peaks across the different immunoprecipitated proteins, we also evaluated the percentage of ASXL2, AML1-ETO and RUNX1 peaks that overlap with promoters (defined as − 2.5 to + 2.5 kb of TSS). The percentage of promoter-overlapping peaks for ASXL2, AML1-ETO and RUNX1 were 9.7, 27 and 29.8% respectively. Although these data suggest that ASXL2 binds fewer TSS than AML1-ETO and RUNX1, ASXL2 binding was strongly enriched at AML1-ETO-binding sites (Fig. 3d–f, Supplementary Fig. 5 and Supplementary Data 2–4). Thus, in the setting of both normal and malignant haematopoietic cells, direct binding sites of ASXL2 strongly overlapped with those of AML1-ETO and RUNX1 (ChIP-seq enrichment for AML1-ETO, RUNX1 and ASXL2 at representative loci are shown in Supplementary Fig. 5a). Consistent with this, motif enrichment analysis of ASXL2-binding sites in these same SKNO−1 cells revealed a strong overlap of ASXL2 binding with ETS as well as AP−1 transcription factors, all established as interacting physically with the AML1-ETO transcriptional complex in human t(8;21) (ref. 33) AML

**Figure 2 | Asxl2 is required for haematopoietic stem cell self-renewal.** (**a**) Schema of serial competitive BMT assays from CD45.2+ *Mx1*-cre control (black), *Mx1*-cre *Asxl1fl/fl* (red), *Mx1*-cre *Asxl2fl/fl* (blue) and *Mx1*-cre *Asxl1fl/fl Asxl2fl/fl* (purple) mice. (**b**) Percentage of CD45.2+ chimerism in the peripheral blood of recipient mice over time (n = 10 mice/genotype) in primary (1°), secondary (2°) and tertiary (3°) BMT. (**c**) Flow cytometric enumeration of CD45.2+ lineage-negative Sca1+ cKit+ (LSK) cells in the BM of *Mx1*-cre control, *Mx1*-cre *Asxl1fl/fl*, *Mx1*-cre *Asxl2fl/fl* and *Mx1*-cre *Asxl1fl/fl & Asxl2fl/fl* mice 14 weeks after polyinosinic:polycytidylic acid (pIpC) injection in competitive transplant. (**d**) Total number of BM MP, LSK cells, LT-HSC, restricted haematopoietic progenitor cell fractions 1 (HPC-1) and 2 (HPC-2) and MPP cells following noncompetitive BMT from the indicated genotypes (as shown in Fig. 1d; n = 5 mice/genotype). (**e**) Percentage of CD45.2+ chimerism in the peripheral blood of recipient mice transplanted with CD45.2*Mx-1* control, *Mx1*-cre *Asxl2fl/WT*, or *Mx1*-cre *Asxl2fl/fl* BM cells over time (n = 10 mice/genotype). (**f**) Flow cytometric enumeration of CD45.2+ LSK cells in the BM of mice from **e** 14 weeks after pIpC injection in competitive transplant. Error bars represent mean ± s.d.; *$P < 0.05$, **$P < 0.001$, ***$P < 0.0002$, ****$P < 0.0001$. P values calculated by ordinary one-way ANOVA test. ANOVA, analysis of variance; LT-HSC, long-term haematopoietic stem cells.

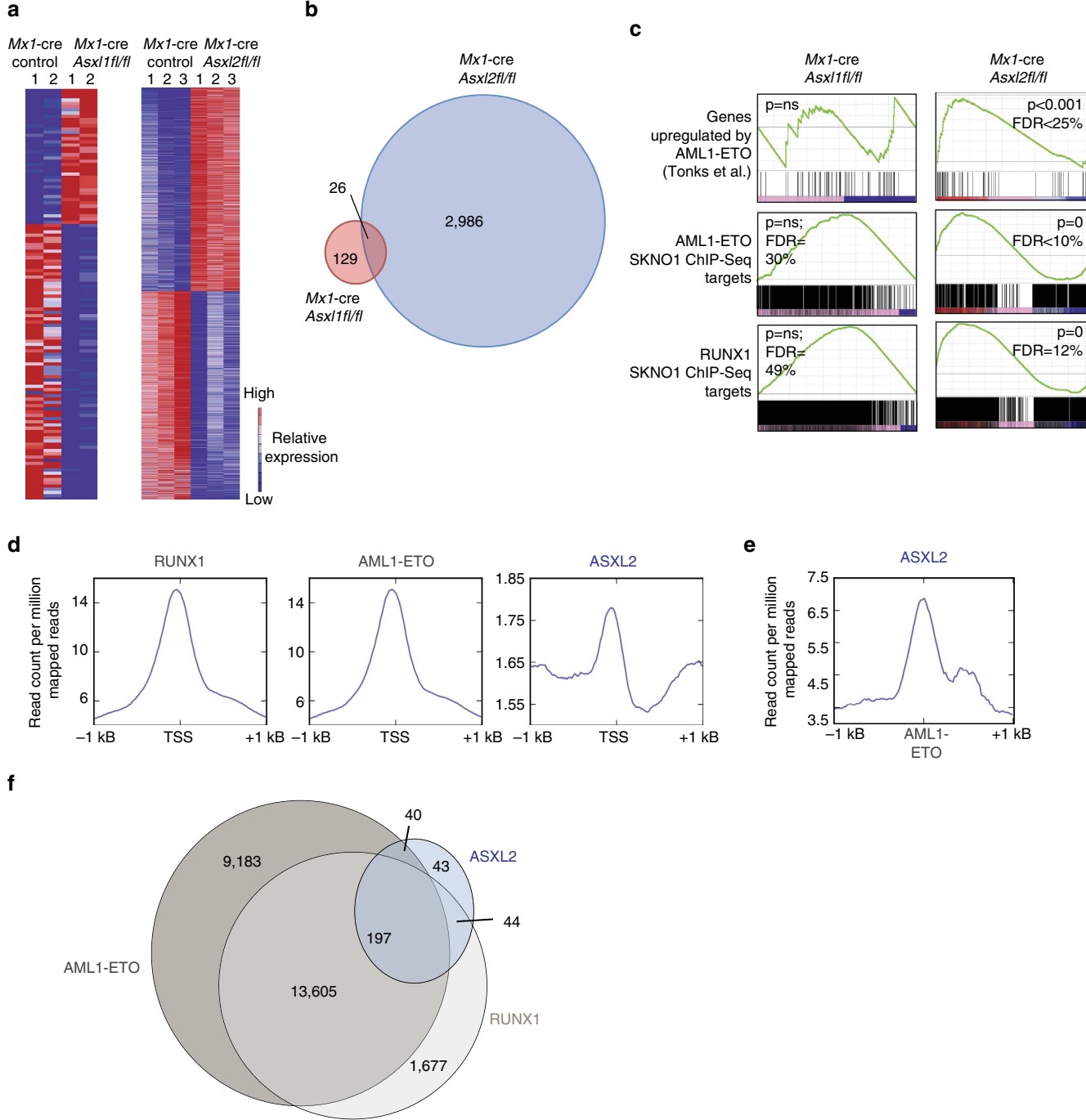

**Figure 3 | ASXL2 target genes overlap with those of RUNX1 and AML1-ETO.** (**a**) Heatmap of significantly differentially expressed genes in LSK cells from *Mx1*-cre *Asx1fl/fl* or *Mx1*-cre *Asxl2fl/fl* mice relative to *Mx1*-control mice (genes shown are those restricted to differential expression with multiple hypothesis adjusted *P* value of *P*<0.05). (**b**) Venn diagram of differentially expressed genes in *Mx1*-cre *Asxl1fl/fl* and *Mx1*-cre *Asxl2fl/fl* LSK cells relative to control (genes restricted to *P* value cutoff described for **a**). (**c**) GSEA of genes enriched in *Mx1*-cre *Asxl2fl/fl* LSK cells (right), with plots of enrichment for same gene sets in *Mx1*-cre *Asxl1fl/fl* shown on left for comparison. (**d**) The average binding intensity of RUNX1, AML1-ETO and ASXL2 centred on TSSs ±1 kB in SKNO-1 cells. (**e**) The average binding intensity of ASXL2 centred on AML1-ETO-binding sites ±1 kB in SKNO-1 cells. (**f**) Overlap of RUNX1, AML1-ETO and ASXL2 binding sites in SKNO1-cells. ChIP-seq reads in **d**–**f** were normalized as read counts per million mapped reads. GSEA, gene set enrichment analysis.

(Supplementary Fig. 5d; Supplementary Data 5 shows all significantly enriched motifs under ASXL2 ChIP-seq peaks, of which ETS and AP−1 motifs represent only a subset).

Given the overlap of ASXL2, RUNX1 and AML1-ETO-binding sites we also sought to examine if there was a direct physical interaction of these proteins. However, co-immunoprecipitation experiments failed to reveal physical interaction of RUNX1,

AML1-ETO or AML1-ETO9a with ASXL2 (Supplementary Fig. 6a,b).

**H3K27me3 loss and alterations at enhancers with ASXL2 loss.** In addition to potential direct transcriptional effects, prior work from us[36] and others[27,37] has identified that ASXL1 alterations

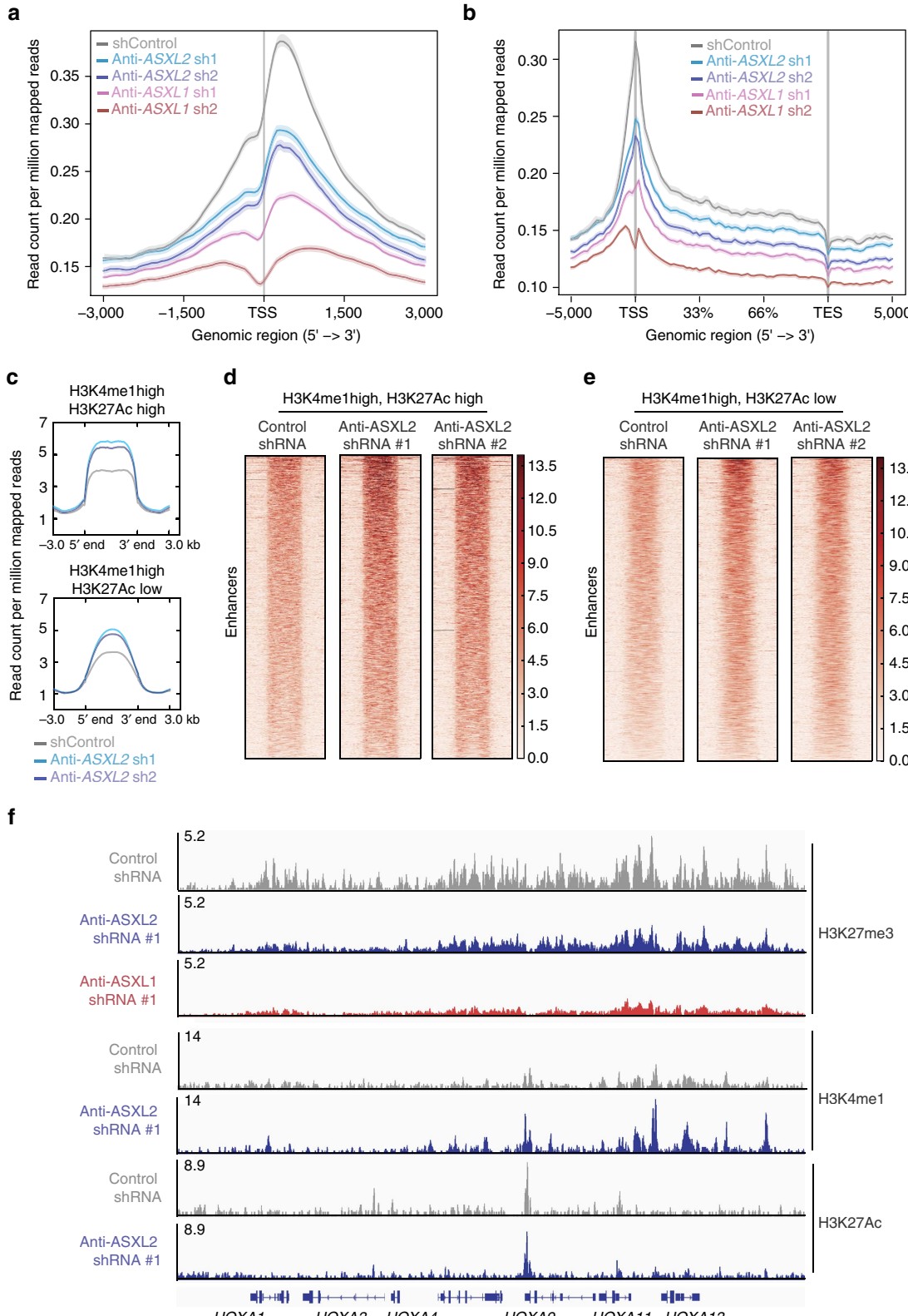

**Figure 4 | Effects of ASXL2 loss on chromatin state of AML1-ETO leukemic cells.** (**a**) Anti-H3K27me3 ChIP-seq signal intensity centred at TSSs ± 3 kB or (**b**) gene-bodies ± 5 kB in SKNO-1 cells treated with control shRNA and two different anti-ASXL1 and ASXL2 shRNAs. TES; transcription end site. (**c**) Mean anti-H3K4me1 ChIP-seq signal density at putative active (top) and poised enhancers (bottom; defined as enhancers marked by H3K4me1 and H3K27Ac signal or just H3K4me1 signal, respectively) in SKNO-1 cells with or without ASXL2 downregulation. ChIP-seq reads in **a**–**c** were normalized and displayed as read counts per million mapped reads. (**d,e**) Data from **c** displayed as a heatmap representation of H3K4me1 ChIP-seq signal at putative (**d**) active and (**e**) poised enhancers in SKNO-1 cells with control shRNA or one of two different ASXL2 shRNAs. (**f**) anti-H3K27me3, H3K4me1 and H3K27Ac ChIP-seq profiles (displayed as read counts per million mapped reads) at the HOXA locus in SKNO-1 cells with control shRNA or anti-ASXL1 or ASXL2 shRNAs.

                                    

are associated with global downregulation of histone H3 lysine 27 trimethylation (H3K27me3). Interestingly, while ASXL1 loss in SKNO-1 cells was associated with clear downregulation of H3K27me3 at TSS and gene bodies, ASXL2-deficient cells had less prominent H3K27me3 loss in comparison (Fig. 4a,b, Supplementary Fig. 6c–e and Supplementary Table 1). Similarly, Asxl1-deficient haematopoietic progenitors exhibited global downregulation of H3K27me3 which was not seen in Asxl2-deficient progenitors based on both western blotting as well as anti-H3K27me3 ChIP-seq (Supplementary Fig. 6f,g). Finally, given prior reports for roles of ASXL1 as well as ASXL2 in PRC2-mediated H3K27me3, we evaluated for a potential physical interaction between ASXL2 or ASXL1 and PRC2 components. Despite evidence of clear physical interaction between ASXL1, ASXL2, and BAP1 (as previously reported[38]) as well as ASXL1 and the core PRC2 member SUZ12 (also previously reported[36]), interaction between ASXL2 and PRC2 members were not evident (Supplementary Fig. 6b).

We therefore next examined the effects of ASXL2 loss on the state of other histone post-translational modifications known to be regulated by Polycomb and/or Trithorax group protein complexes. While ASXL2 loss had no clear effect on histone H3 lysine 4 trimethylation (H3K4me3) at TSS (Supplementary Fig. 6h), ASXL2 loss was associated with increases in H3K27ac and H3K4me1 signal at putative active enhancers (marked by both H3K27ac and H3K4me1) as well as poised enhancers (marked by H3K4me1 without H3K27ac) as seen by anti-H3K4me1 and anti-H3K27Ac ChIP-seq following ASXL2 depletion using two independent shRNAs targeting ASXL2 (Fig. 4c–f). To correlate alterations in histone modifications following ASXL2 loss in AML1-ETO leukemic with genes differentially expressed by ASXL2 loss we performed RNA-seq analysis of SKNO1 cells treated with control shRNA or 2 different anti-ASXL2 shRNAs, each in biological triplicate (Supplementary Fig. 7a and Supplementary Data 6–7). Interestingly, this identified a number of genes known to promote leukemogenesis (either alone or in the context of AML1-ETO leukemia) as differentially expressed by ASXL2 loss. These include downregulation of TET2 as well as NOTCH2 with ASXL2 loss in human AML1-ETO-expressing cells, downregulation of which have been previously shown to functionally promote myeloid leukemogenesis when altered in expression[24,39,40]. Next, we utilized these RNA-seq data to evaluate the overlap of differentially expressed genes with direct targets of ASXL2, RUNX1, and AML1-ETO as well as alterations in H3K27me3, H3K4me1 and H3K27Ac. This revealed that ∼90% of genes dysregulated in expression with ASXL2 loss are direct targets of RUNX1 or RUNX1-ETO (Supplementary Fig. 7b). There was no clear correlation between the loss of H3K27me3 upon ASXL2 knockdown and alterations in gene expression (Supplementary Fig. 7c). Similarly, genes within 100 kB of H3K4me1 or H3K27Ac peaks in shControl cells did not experience consistent alterations in gene expression following ASXL2 loss (although this may be explained by the fact that these are putative enhancers that do not necessarily regulate the expression of genes located in 100 kB proximity of these genes; Supplementary Fig. 7d,e).

To evaluate if the histone modifications seen with ASXL2 loss were altered at ASXL2 or AML1-ETO-binding sites specifically, we analysed the H3K27me3 and H3K4me1 abundance at ASXL2 peaks as well as AML1-ETO peaks in SKNO-1 cells treated with control shRNA or one of 2 different anti-ASXL2 shRNAs. Although changes in H3K27me3 were more prominent at ASXL2 or AML1-ETO peaks than at random control regions of the genome outside of these peaks (Supplementary Fig. 7f,g), H3K27me3 changes were prominent across TSS and gene bodies (Fig. 4a,b). Moreover, there was no clear correlation between

changes in H3K4me1 and sites of ASXL2 binding (Supplementary Fig. 7h) suggesting that alterations in H3K27me3 and H3K4me1 did not have a specific relation to ASXL2-binding sites.

**Asxl2 loss promotes AML1-ETO-mediated leukemogenesis.** Genetic analyses of patients with t(8;21) AML and extensive animal modelling of the haematopoietic effects of the AML1-ETO translocation have identified that AML1-ETO is expressed in early HSCs and increases self-renewal but is not sufficient for leukemogenesis alone[20–22]. Substantial effort has therefore been spent to identify genetic events that collaborate with AML1-ETO to promote leukemogenesis. Recent genomic analyses of AML1-ETO AML patients have identified that the average variant allele frequency of mutations in ASXL2 are significantly higher than those of other mutations frequently co-existing with AML1-ETO such as c-KIT, FLT3 and N/KRAS mutations[25]. Consistent with this, unsupervised hierarchical clustering of differentially expressed genes amongst AML patients with the AML1-ETO translocation revealed that ASXL2-mutant AML1-ETO samples form a distinct transcriptional subset of AML1-ETO AML, an effect not seen with other common genetic alterations in AML1-ETO AML (Fig. 5a).

Given the above human genetic data we next sought to examine the contribution of ASXL2 loss to AML1-ETO-mediated leukemia. We performed murine retroviral BMT assays over-expressing full-length AML1-ETO or the AML1-ETO9A (AE9a) splice isoform[41] linked to GFP in BM cells from Asxl2-deficient or control mice followed by transplantation into WT C57BL/6 recipients (Supplementary Fig. 8a,b). Converse to the failure of HSC self-renewal with deletion of Asxl2 in normal haemato-poiesis, loss of Asxl2 in the setting of either full-length AML1-ETO or AE9a overexpression consistently led to hastened death due to shortened latency of leukemogenicity compared with AML1-ETO or AE9a overexpression in an Asxl2 WT background (Fig. 5b,c). Consistent with this, mice transplanted with AE9a/Asxl2-null leukemias had a higher frequency of GFP/c-Kit double-positive cells in both peripheral blood and BM at the time of death (Supplementary Fig. 8c).

Next to study leukemogenesis mediated by AML1-ETO and ASXL2 loss in the genetic configuration most representative of human AML, we crossed mice bearing expression of the Aml1-Eto fusion from the endogenous locus of Aml1 (Aml1-Eto conditional knockin mice[23]) to Mx1-cre Asxl2fl/fl mice to generate Mx1-cre Aml1-Eto Asxl2fl/WT and Mx1-cre Aml1-Eto Asxl2 WT mice (Supplementary Fig. 9a–c). Heterozygous loss of Asxl2 collaborated with expression of endogenous Aml1-Eto to promote leukemo-genesis (Fig. 5d). Moreover, Mx1-cre Aml1-Eto Asxl2fl/WT mice developed prominent extramedullary leukemic infiltration in soft tissues, a clinical characteristic often seen in patients with this subtype of AML[42,43] (Fig. 5e–g and Supplementary Fig. 9d). Finally, to model the effects of coexistent oncogenic RAS mutations, which are present in a subset of AML1-ETO/ASXL2-mutant AML patients[25], expression of NRASG12D in Mx1-cre Aml1-Eto Asxl2fl/WT and Mx1-cre Aml1-Eto Asxl2 WT BM backgrounds resulted in further collaborative effects in the AML1-ETO/ASXL2 haploinsufficient background (Fig. 5d and Supplementary Fig. 9e–h). Mx1-cre Aml1-Eto Asxl2fl/WT mice expressing NRASG12D developed worsened anaemia and thrombocytopenia as well as greater circulating GFP/c-Kit double-positive cells and hastened death compared with counterpart Mx1-cre Aml1-Eto Asxl2 WT mice expressing NRASG12D.

Given the alterations of enhancers in human AML1-ETO cells with ASXL2 loss, we next examined the chromatin state of AE9a leukemias with or without Asxl2 expression by transposase-accessible chromatin sequencing (ATAC-Seq).

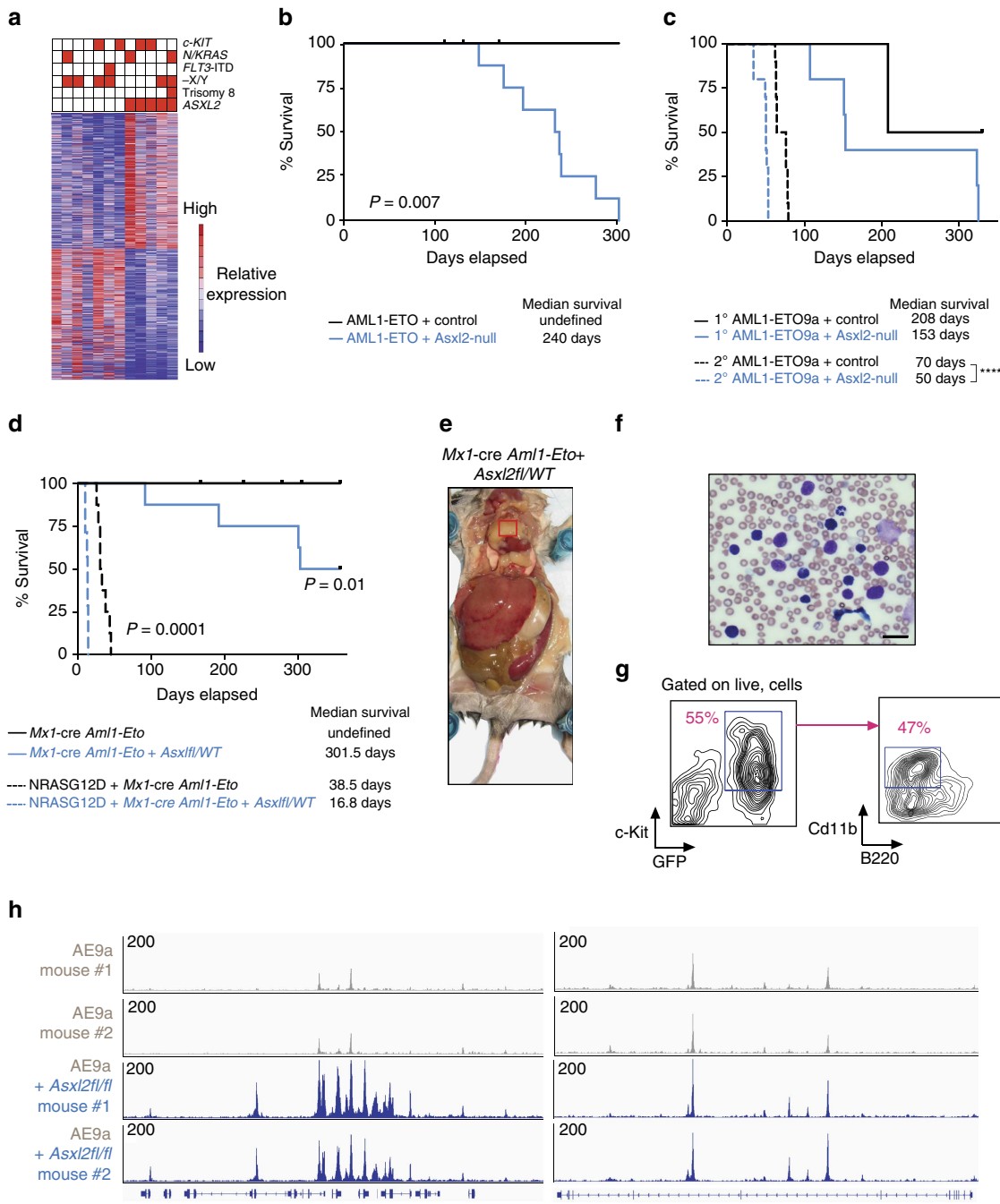

**Figure 5 | Asxl2 loss promotes AML1-ETO-mediated leukemogenesis.** (**a**) Heatmap displaying unsupervised hierarchical clustering of differentially expressed genes amongst AML1-ETO patient BM samples at diagnosis. Genes mutated in each sample is shown above the heatmap. (**b,c**) Kaplan–Meier survival curves of recipient mice with overexpression of (**b**) AML1-ETO or (**c**) AML1-ETO9a (AE9a) in *Asxl2* wild-type (WT) or *Asxl2*-null BM in primary and secondary transplantation assays ($n = 10$ mice for each genotype and each round of transplantation). (**d**) Kaplan–Meier survival curves of recipient mice transplanted with *Mx1-cre Aml1-Eto* control or *Mx1-cre Aml1-Eto Asxl2fl/WT* BM with or without NRASG12D overexpression ($n = 8$ mice for each genotype). (**e,f**) Photograph of representative recipient of *Mx1-cre Aml1-Eto Asxl2fl/WT* mouse at necropsy revealing (**e**) extramedullary leukemic mass over heart (red box), hepatosplenomegaly and (**f**) circulating blasts (scale bars, 25 μm, Wright-Giemsa stain). (**g**) Flow cytometric analysis of extramedullary AML from representative *Mx1-cre Aml1-Eto Asxl2fl/WT* mouse (tissue from red box in **e**. (**h**) ATAC-Seq profiles (displayed as read counts per million mapped reads) at the *HoxA* (left) and *Meis1* (right) loci in GFP+/c-Kit+ cells from two different AE9a/*Mx1*-cre control (grey) and AE9a/*Mx1*-cre *Asxl2fl/fl* (blue) mice. P values calculated by log-rank test.

Although global chromatin accessibility as assessed by ATAC-seq of GFP/c-Kit double-positive cells from two independent AE9a/*Mx1*-cre control and 2 AE9a/*Mx1*-cre *Asxl2fl/fl* mice did not reveal substantial global differences between groups (Supplementary Fig. 10), there was a striking increase in chromatin accessibility at the *HoxA* and *Meis1* loci of AE9a/*Mx1*-cre *Asxl2fl/fl* mice compared to AE9a control mice with Asxl2 intact (Fig. 5h). These data suggest that Asxl2 loss

promotes AML1-ETO leukemogenesis by altering enhancer accessibility at key leukemogenic loci.

## Discussion

Mutations in genes encoding Polycomb[4,5,26,27,44] and Trithorax[45–48] group family members and their associated proteins are amongst the most common genetic alterations in haematological malignancies. Mutations in the polycomb-associated protein *ASXL1* are present across the myeloid neoplasms and ubiquitously associated with adverse clinical outcome in every subtype of myeloid leukemia[7,8,49–51]. Consistent with the frequency of genetic alterations of *ASXL1* in patients, deletion or mutation of *Asxl1 in vivo* results in myelodysplastic syndrome-like disease in mice[26,27]. However, the biological effects of *Asxl1* loss or mutation expression in the haematopoietic system are modest[26,27] and this has led to the hypothesis that the function of *ASXL2* may be partially redundant with that of *ASXL1* and able to compensate for the absence of ASXL1 (ref. 52). At the same time, it has been proposed that ASXL1 and ASXL2 may even have opposing biological functions[13]. We therefore set out to understand the biological and transcriptomic effects of Asxl2 loss in the haematopoietic system and to compare them to Asxl1 loss. We identify here that Asxl2 is required for haematopoiesis in a gene dosage-dependent manner and has non-overlapping biological functions with Asxl1. Future efforts to compare the direct binding targets of ASXL1 versus ASXL2 in normal haematopoietic cells, the compendium of proteins interacting with ASXL1 versus ASXL2, and distinct domains between the two proteins will be critical in further understanding the mechanistic basis for the differing effects of ASXL1 versus ASXL2 loss.

Consistent with the fact that Asxl2 loss resulted in greater defects in haematopoiesis than Asxl1 loss, Asxl2 deletion was also associated with a larger number of differentially expressed genes in HSPCs than deletion of Asxl1. Interestingly, a substantial number of genes dysregulated by Asxl2 loss appeared to be direct targets of RUNX1 and AML1-ETO. Recent genomic analysis has delineated that mutations in *ASXL2* are restricted to AML1-ETO AML[14–16,25]. This is in stark contrast to mutations in *ASXL1*, which are present throughout myeloid neoplasms and also enriched in clonal haematopoiesis in normal individuals[17–19]. These human genetic data suggest that ASXL2 may promote leukemogenesis or increased self-renewal only in the setting of expression of AML1-ETO and not in the setting of normal HSPCs. Consistent with this, we identified that ASXL2 loss promoted leukemogenesis in AML1-ETO-expressing HSPCs but did not promote the self-renewal of normal haematopoietic stem cells. Loss of Asxl2 even accelerated leukemogenesis of AE9a, a splice variant of AML1-ETO whose overexpression alone is leukemogenic[41]. Although the shorter latency of AE9a/Asxl2-null leukemias over AML1-ETO/Asxl2-null leukemias is likely to reflect the known enhanced leukemogenicity of AE9a over AML1-ETO (ref. 41), future efforts to decipher if there is a specific functional collaboration between AE9a and ASXL2 loss may be enlightening. Similarly it will be interesting to compare the effects of Asxl1 versus Asxl2 loss in the setting of AML1-ETO or AE9a expression.

Asx proteins were originally identified as being required for assisting Polycomb and Trithorax group proteins in maintaining homeotic gene expression and silencing[53,54]. Here, ASXL2 loss in the context of AML1-ETO AML cells was associated with loss of repressive H3K27me3 modifications as well as increased H3K27ac and H3K4me1 signal at enhancers. Currently the molecular mechanisms by which ASXL2 loss results in these changes in chromatin state is not known. Recent work has suggested that the plant homeofinger domain (PHD) of ASXL2 may bind to and influence monomethylation at H3K4 (ref. 55), a mark associated with functional enhancers[56–58]. This potential function of the ASXL2 PHD domain may provide a mechanistic explanation for the observation here of alterations of enhancers with loss of ASXL2. However, further confirmation of the role of the ASXL2 PHD domain as well as functional comparison to the PHD domain of ASXL1 is needed.

As noted earlier, despite being among the common fusions in AML patients, the AML1-ETO fusion protein by itself is not sufficient to promote overt leukemia[20–24]. The fact that loss of even a single copy of Asxl2 promoted leukemogenesis mediated by endogenous Aml1-Eto expression strongly supports the concept that Asxl2 is a haploinsufficient tumour suppressor in the context of this subtype of AML. Genes dysregulated by ASXL2 loss largely overlapped with those of RUNX1 and AML1-ETO and were associated with alterations in a number of genes previously shown to promote myeloid leukemias. Moreover, in both human and mouse AML1-ETO-expressing cells, ASXL2 loss was associated with increased chromatin accessibility at the *HoxA* locus, expression of which is well established as promoting myeloid leukemogenesis[59–62]. Although the overlap of ASXL2 dysregulated genes and gene targets of AML1-ETO and RUNX1 suggest potential physical overlap of ASXL2, AML1-ETO and RUNX1, no physical interaction between these 3 proteins were detected here. Future efforts to map Asxl2 binding sites in the context of normal HSPCs compared with those of Runx1 may be helpful in further elucidating the basis for the overlap in gene expression between Asxl2, RUNX1, and AML1-ETO.

Given the paucity of genetically accurate models for AML1-ETO AML based on co-occurring genetic lesions, the models presented in this study may be critical for understanding AML1-ETO disease pathogenesis further in addition to providing conditional alleles to define the role of ASXL2 outside of the haematopoietic system. Overall, these data highlight Asxl2 as critical for normal haematopoiesis as well as a novel haploinsufficient tumour suppressor in leukemia.

## Methods

**Human subjects.** RNA extracted from peripheral blood or BM mononuclear cells from adult t(8;21) AML patients was utilzed for RNA sequencing and included 7 *ASXL1/2* wild-type and 5 *ASXL2* mutants from the CBF-2006 trial (EudraCT #2006 005163-26; ClinicalTrials.gov identifier #NCT00428558). The study was approved by the ethics committee of Nimes University Hospital and by the Institutional Review Board of the French Regulatory Agency and conducted in accordance to the Declaration of Helsinki protocol. Molecular and cytogenetic analyses were performed as described previously[15].

**Animals.** All animals were housed at Memorial Sloan Kettering Cancer Center (MSKCC). All animal procedures were conducted in accordance with the Guidelines for the Care and Use of Laboratory Animals and were approved by the Institutional Animal Care and Use Committees at MSKCC.

**Generation of Asxl2 and Asxl1/2 conditional knockout mice.** The *Asxl2* allele was deleted by targeting exon 11. Two LoxP sites flanking exon 11 and a Frt-flanked neomycin selection cassette were inserted in the upstream intron (Fig. 1a). Ten micrograms of the targeting vector was linearized by NotI and then transfected by electroporation of C57Bl/6 (B6) embryonic stem cells (ES). After selection with G418, surviving clones were expanded for PCR analysis to identify recombinant ES clones. Positive clones identified by PCR were confirmed independently by Southern blotting analysis (Supplementary Fig. 1c). DNA was digested with Mfe l, and electrophoretically separated on a 0.8% agarose gel. After transfer to a nylon membrane, the digested DNA was hybridized with a probe targeted against the 3′-external region. DNA from C57Bl/6 (B6) mouse strain was used as a wild-type control. The *Asxl2fl* mouse line will be available from The Jackson Laboratory as JAX#030338.

The generated mice (*Asxl2fl/fl*) were initially crossed to a germline Flp deleter (Jackson Laboratory) to eliminate the neomycin cassette, and subsequently to the IFN-α inducible *Mx1-cre* (Jackson Laboratory) strain. *Asxl2fl/fl, Asxl2fl/WT* and *Asxl2WT/WT* littermate mice were genotyped by PCR with primers Asxl2-F (5′-GCAGGCTCTCTACAAACTCAGTTC-3′) and Asxl2-R (5′-CAACATC

GATATTGCTACTGA TAA AGTGAA-3′) using the following parameters: 94 °C for 2 min, followed by 35 cycles of 94 °C for 30 s, 62 °C for 30 s and 72 °C for 1 min, and then 72 °C for 5 min. The wild-type allele was detected as a band at 330 bp, whereas the floxed allele was detected as a band of 517 bp. Excision after cre-mediated recombination was confirmed by western blot and quantitative PCR with reverse transcription.

**Cell culture.** SKNO-1 cells were purchased from DSMZ (Braunschweig, Germany) and cultured in RPMI 1640 with 20% FCS, 1% penicillin-streptomycin, and 10 ng ml$^{-1}$ GM-CSF (BD Biosciences).

**Transfection and retrovirus production.** cDNAs encoding AML1-ETO9a was digested with NotI and cloned into the NotI multiple cloning site of the MSCV-IRES-BEX plasmid (pBEX), upstream of the IRES and blue-excited GFP motifs[24]. To produce retrovirus capable of expressing AML1-ETO9a, 293T cells were cotransfected with a pBEX plasmid and MCV-Ecopac using X-treme GENE 9 (Roche) transfection reagent.

**Mouse bone marrow transplantation assays.** Freshly dissected femora and tibiae were isolated from Mx1-cre control, Mx1-cre Asxl1fl/fl, Mx1-cre Asxl2fl/fl, Mx1-cre Asxl1fl/fl Asxl2fl/fl mice. BM was flushed with a 3-cc insulin syringe into cold PBS (without Ca$^{2+}$ and Mg$^{2+}$) supplemented with 2% bovine serum albumin to generate single cell suspensions. BM cells were spun at 252 g for 5 min by centrifugation and red blood cells (RBCs) were lysed in ammonium chloride-potassium bicarbonate lysis (ACK) buffer for 5 min on ice. After centrifugation, cells were resuspended in PBS/2% BSA, filtered through a 40 μM cell strainer. For competitive transplantation experiments, $0.5 \times 10^6$ total BM cells from Mx1-cre control, Mx1-cre Asxl1fl/fl, Mx1-cre Asxl2fl/fl, Mx1-cre Asxl1fl/fl Asxl2fl/fl CD45.2 + mice were mixed with $0.5 \times 10^6$ wild-type CD45.1 + support BM and transplanted via tail vein injection into 6-week-old lethally irradiated $(2 \times 450\text{cGy})$ CD45.1 + recipient mice. To activate the conditional alleles, mice were treated with 3 doses of pIpC (12 mg kg$^{-1}$ day$^{-1}$; GE Healthcare) via intra-peritoneal injection 4 weeks post transplant. Peripheral blood chimerism was assessed every 4 weeks by flow cytometry. For noncompetitive transplantation experiments, $1 \times 10^6$ total BM cells from from Mx1-cre control, Mx1-cre Asxl1fl/fl, Mx1-cre Asxl2fl/fl, Mx1-cre Asxl1fl/fl Asxl2fl/fl CD45.2 + mice were injected into lethally irradiated $(2 \times 450\text{cGy})$ CD45.1 + recipient mice. Peripheral blood chimerism was assessed as described in competitive transplantation experiments. Additionally, for each bleeding whole blood cell counts were measured on an automated blood analyser.

For AML1-ETO and AML1-ETO9a retroviral primary BMT experiments, donor cells from Mx1-cre control or Mx1-cre Asxl2fl/fl mice were treated with a single dose of 5-fluorouracil (150 mg kg$^{-1}$) followed by BM harvest from the femora, tibiae and hip bones 6 days later. RBCs were removed by ACK lysis buffer, and nucleated BM cells were transduced with viral supernatants containing MSCV-AML1-ETO9a-IRES-BEX for 2 days followed by injection of 100,000 pBEX + cells per recipient mouse via tail vein injection into lethally irradiated $(2 \times 450\text{cGy})$ CD45.1 recipient mice associated with 500,000 wild-type CD45.1 + support BM. For secondary transplantation experiments, 6-week-old, sub-lethally irradiated (450cGy) C57/BL6 recipient mice were injected with $1 \times 10^6$ primary AML1-ETO9a leukemic cells per recipient.

**Methylcellulose colony assays.** Megakaryocyte colonies were processed using STEMCELL Technologies' protocol for MegaCult-C products for Mouse CFU-Mk Assays. Briefly, LSK cells were sorted by flow cytometry and were cultured for 10 days in a double chamber culture slide (5,000 cells/chamber) with serum-free medium containing cytokines (rhThrombopoietin 50 ng ml$^{-1}$, rhIL-6 20 ng ml$^{-1}$, rmIL-3 10 n ml$^{-1}$). After dehydration and fixation the cells were stained for Megakaryocytes Acetylcholinesterase activity. CFU-Mk that were acetylcholinesterase positive and contained at least 3 megakaryocytes were counted.

**Antibodies.** Cell populations were analysed using a FACSFortessa (Becton Dickinson) and sorted with a FACSAria II instrument (Becton Dickinson). All FACS antibodies were purchased from BD Pharmingen, BioLegend or eBioscience. We used the following antibodies: c-Kit (2B8, Cat# 105808, 1:100), Sca-1 (D7, Cat# 108114, 1:100), B220 (RA3-6B2, Cat# 103222, 1:200), Mac-1/CD11b (M1/70, Cat# 101206, 1:200), Gr-1 (RB6-8C5, Cat# 108412, 1:200), NK1.1 (PK136, Cat# 108724, 1:100), Ter119 (TER-119, Cat# 116223, 1:100), IL-7 R A7R34, Cat# 135014, 1:100), CD3 (17A2, Cat# 100222, 1:100), CD4 (RM4-5, Cat# 100526, 1:100), CD8 (53-6.7, Cat# 100712, 1:100), CD16/32 (93, Cat# 56-0161-82, 1:100), CD34 (RAM34, Cat# 11-0341-85, 1:100), CD41 (MWReg30, Cat# 133916, 1:100), CD45.1 (A20, Cat# 110738, 1:100), CD45.2 (104, Cat# 109822, 1:100), CD48 (HM48-1, Cat# 103422, 1:100), CD71 (RI7217.1.4, Cat# 12-1501-82,1:100), and CD150 (mShad150, Cat# 12-1502-82,1:100).

The following antibodies were used for Western Blot analysis: anti-ASXL2 (Bethyl laboratories A302-037A, 1:2,000), anti-ASXL1 (Santa-Cruz sc-85283, 1:200), anti-beta-Actin (Sigma-Aldrich A5441, 1:1,000) and anti-Tubulin (Sigma-Aldrich T6199, 1:1000).

Antibodies used for ChIP include anti-H3K4me3 (Cell Signaling 9751S, 1:50), anti-H3K27ac (Abcam ab4729, 2 μg for 25 μg of chromatin) and anti-H3K4me1 (Abcam ab8895, 2 μg for 25 μg of chromatin), anti-H3K27me3 (Millipore 07–449, 2 μg for 25 μg of chromatin), anti-ASXL1 (Santa Cruz sc-98302, 1:50) and anti-ASXL2 (Bethyl A302–037A, 2 μg for 25 μg of chromatin).

**Histone extraction.** Histones were extracted using the Active Motif Histone Extraction Minikit (Active Motif 40026).

**ChIP-Seq and motif analysis.** For ChIP-seq in murine cells 10 million BM c-Kit + cells (isolated with anti–mouse CD117 microbeads (Miltenyi Biotec) were used. For ChIP in SKNO-1 cells, 10 millions of GFP + cell were used. Briefly, cells were fixed in a 1% methanol-free formaldehyde solution and then resuspended in sodium dodecyl sulfate (SDS) lysis buffer. Lysates were sonicated in an E220 focused-ultrasonicator (Covaris) to a desired fragment size distribution of 100–500 base pairs. IP reactions were performed with the indicated antibodies, each on approximately 500,000 cells[63]. ChIP assays were processed on an SX-8G IP-STAR Compact Automated System (Diagenode) using a direct ChIP protocol as described elsewhere[64]. Eluted chromatin fragments were then de-crosslinked and the DNA fragments purified using Agencourt AMPure XP beads (Beckman Coulter).

Barcoded libraries were prepared from the ChIP-enriched and input DNA using a NEBNext ChIP-seq Library Prep Master Mix Set for Illumina (New England BioLabs) and TruSeq Adaptors (Illumina) according to the manufacturer's instructions on an SX-8G IP-STAR Compact Automated System (Diagenode). Phusion High-Fidelity DNA Polymerase (New England BioLabs) and TruSeq PCR Primers (Illumina) were used to amplify the libraries, which were then purified to remove adaptor dimers using AMPure XP beads and multiplexed on the HiSeq 2000 (Illumina). Previously published anti-RUNX1 and AML1-ETO ChIP-seq were downloaded from the NCBI Gene Expression Omnibus (GEO;http://www.ncbi.nlm.nih.gov/geo/) under accession no. GSE23730 (ref. 30).

Raw ChIP-seq data were analysed using Basepair software (http://www.basepairtech.com/) with pipelines including the following steps: the raw fastq data were trimmed using trim_galore to remove low-quality ends from reads (quality<15) and adapter sequences. The trimmed data was aligned using Bowtie2 (ref. 65) to UCSC genome assembly hg19 (for human samples) or mm9 (for mouse samples). Duplicate reads were removed using Picard and bigwig files were created for visualization. Peaks were identified with Macs1.4 (ref. 66) and transcription factor binding motifs were detected with Homer. Peaks overlapping with Satellite repeat regions were discarded and remaining filtered peaks were annotated using custom scripts based on UCSC refFlat data[67], where peaks between − 2500 bp to 2500 bp of a transcription start site were marked as Promoter, the overlapping gene body were marked as Genebody and the rest were marked as Intergenic. For intergenic peaks, a gene was considered a target if it was within 1 Mb of the peak. All ChIP-seq reads were normalized and displayed as read counts per million mapped reads.

**ATAC-seq.** ATAC-seq was performed as previously described[68]. For each sample, nuclear extracts were prepared from 50,000 cells, and incubated with 2.5 μl of transposase (Illumina) in a 50 μl reaction for 30 min at 37 °C. After purification of transposase-fragmented DNA, the library was amplified by PCR and subjected to high-throughput sequencing using the HiSeq 2000 platform (Illumina). ATAC-seq reads were normalized and displayed as read counts per million mapped reads.

**mRNA, sequencing.** RNA-Seq was conducted with three biological replicates from each group. Genetic phenotyping experiments were replicated three times independently. RNA was extracted from sorted mouse cell populations using Qiagen RNeasy columns. poly(A)-selected, unstranded Illumina libraries were prepared with a modified TruSeq protocol. 0.5X AMPure XP beads were added to the sample library to select for fragments <400 bp, followed by 1X beads to select for fragments >100 bp. These fragments were then amplified with PCR (15 cycles) and separated by gel electrophoresis (2% agarose). 300 bp DNA fragments were isolated and sequenced on the Illumina HiSeq 2000 (~100 M 101 bp reads per sample). Differential expression tests were performed using the Cuffdiff module of Cufflinks with RefSeq genes provided as an annotation (− N,− and − M options engaged). We considered genes that had a P < 0.05 to be significantly different between genotypes.

**Statistical analyses.** Data were analysed using GraphPad Prism 7 software. A 2-tailed Student's T test was performed in 2-group comparisons. When comparing multiple groups, one-way analysis of variance followed by Tukey's post-hoc test was performed. *P < 0.05; **P < 0.01; **P < 0.001. A log-ranked (Mantel–Cox) test was used to calculate statistical significance in Kaplan–Meier survival studies.

**Histological analyses.** Mice were sacrificed and autopsied, and then dissected tissue samples were fixed for 24 h in 4% paraformaldehyde, dehydrated, and embedded in paraffin. Paraffin blocks were sectioned at 4 μm and stained with

haematoxylin and eosin (H&E). Images were acquired using an Axio Observer A1 microscope (Carl Zeiss).

**Analysis of protein stability.** HEK293T cells were transfected with full-length wild-type ASXL2 cDNA or ASXL2 cDNAs bearing ASXL2 mutations (p.E1287X, p.T740NfsX16)) followed by treatment with DMSO or cycloheximide (100 ug ml$^{-1}$). After 12 h exposure cells were harvested and results were analysed by western blotting using ASXL2 antibody. The Quantification of the 3estern bands was done using ImageJ Software.

**Data availability.** All ChIP-, ATAC-, and RNA-seq data from this study are deposited in the Gene Expression Omnibus (GEO) under accession number GSE84365. All other remaining data are available within the Article and Supplementary Files, or available from the authors upon request.

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

## Acknowledgements

J.-B.M. is supported by grants from the Fondation de France and Philippe Foundation. AP is a Mildred-Scheel Postdoctoral Research Fellow of the Deutsche Krebshilfe e.V. (number 111354). D.I. is supported by Post-doctoral Fellowships for Research Abroad from the Japan Society for the Promotion of Science, the YASUDA Medical Foundation, and the Kanae Foundation for the Promotion of Medical Science. EK is supported by the Worldwide Cancer Research Fund. SC-WL is supported by a Leukemia and Lymphoma Society (LLS) Special Fellow Award. BHD is supported by the American Society of Hematology (ASH). AY is supported by the Aplastic Anemia and MDS Research Foundation. OAW is supported by grants from the Edward P. Evans Foundation, the Dept. of Defense Bone Marrow Failure Research Program (BM150092 and W81XWH-12-1-0041), NIH/NHLBI (R01 HL128239), an NIH K08 Clinical Investigator Award (1K08CA160647-01), the Josie Robertson Investigator Program, a Damon Runyon Clinical Investigator Award, an award from the Starr Foundation (I8-A8-075), the Leukemia and Lymphoma Society, and the Pershing Square Sohn Cancer Research Alliance.

## Author contributions

J.-B.M., A.P., D.I., and O.A.-W. designed the research studies, J.-B.M., A.P., D.I., E.K., S.C.-W.L., Y.R.C., H.C., X.J.Z., A.Y. and A.K. conducted experiments, J.-B.M., D.I., E.K., S.C.-W.L., Y.R.C., H.C. acquired data, J.-B.M., A.P., D.I., E.K., S.C.-W.L., B.H.D., R.K., A.S., and O.A.-W. Analysed data, N.D., E.S. and C.P. provided reagents, and J.-B.M., A.P., D.I., and O.A.-W. wrote the manuscript.

## Additional information

**Competing interests:** A.S. is an employee of Basepair Inc. All other authors have no competing financial interests.

