## [Peer Review File · Nature Communications]

Reviewers' comments:

Reviewer #1 (Remarks to the Author): Expert in leukaemia and epigenetics

In this study, the authors showed that ASXL2 is required for normal hematopoiesis, has functions distinct from ASXL1 and serves as a haploinsufficient tumor suppressor. While *Asxl2* was required for normal hematopoietic stem cell self-renewal, *Asxl2* loss promoted leukemogenesis by AML1-ETO. Interestingly, ASXL2 target genes strongly overlapped with those of RUNX1 and AML1-ETO and ASXL2 loss was associated with mild reduction in H3K27me3 levels and increased enhancer chromatin marks, leading to higher chromatin accessibility at putative enhancers of key leukemogenic loci. These data correspond well to the findings that ASXL2 mutations are frequently observed in AML patients with AML1-ETO and provide a mechanistic insight into the pathogenesis of AML M2 with ASXL2 mutations.

The study is well designed and the data are solid. This is a paper with high impacts uncovering for the first time the role of ASXL2 in hematopoiesis and leukemogenesis. To improve the manuscript, I have several points as follows.

Major:

1. In Figure 3f, did the author compare with ASX1 targets? Coexistence of ASXL2 with RUNX1 and AML-ETO is intriguing. Is there any physical interaction between ASXL2 and RUNX1 or AML-ETO? This is very important point. Can the author provide any molecular mechanisms for coexistence?
2. Upon ASXL2 KD in SKO1 cells, how do the overlapping targets of ASXL2 and AML-ETO change in expression? What about in *Asxl2* KO HSPCs?
3. In Figure 4, did the H3K27me3 levels and enhancer histone marks change globally or specifically at the ASXL2 or AML-ETO targets? This point could be clarified by analyzing the histone changes at ASXL2/AML-ETO targets compared with non-targets. Did H3K27me3 also change in primary *Asxl2* KO HSPCs? What is the mechanism underlying this phenomenon? Does ASXL2 interact with PRC2 as ASXL1 does?
4. Please show the global changes in H3K27me3, H3K4me1, H3K27ac in *Asxl2* KO HSPCs by Western blot analysis.
5. Please show the correlation between histone changes (H3K27me3 at promoters and enhancer marks) and gene expression. Is there clear correlation?

Minor:

1. Do *Asxl2* KO show dysplasia of myeloid cells? Are the hematopoietic defects observed in *Asxl2* KO compatible with MDS? What is the cause of early death of the mice?
2. In Figure 5c, AML1-ETO9a is highly potent. Is there any functional relationship between AML1-ETO9a and ASXL2? Does AML1-ETO9a antagonize ASXL2 function or recruitment to its targets? If anything known, please discuss?
3. Does dysfunction of ASXL1 also collaborated with AML-ETO in leukemogenesis? Please discuss.

Reviewer #2 (Remarks to the Author): Expert in haematopoiesis

The work described in this manuscript by Micol et al. is based on the initial observation that ASXL2 mutations found in AML with a t(8;21) translocation lead to a decreased protein level of ASXL2. To study the consequences of loss, or decreased levels, of ASXL2 the authors use a conditional model of MOZ deletion using the MX-cre ASXL2 Flox system and show that ASXL2 is crucial for the maintenance of HSCs and progenitors. Moreover, they show that loss of ASXL2 cooperates with AML1-ETO for transformation and that ASXL2 binds to AML1-ETO target genes.

Overall, the experiments were well performed and the results are interesting. The manuscript could be improved by addressing the comments below.

Major comments

1- More data on the primary Mx-cre ASXL2 model, without transplantation, should be provided, especially on the numbers of the different bone marrow progenitor populations (HSC and GMP in particular) and mature populations.

2- Page 6: "ASXL2 is haploinsufficient with regards to normal HSPC function and number" – the authors show that HSPC function is affected but it is unclear if they can say numbers are reduced as they don't show data for non-transplanted transgenic ASXL2 het mice

3- In the figure 5, the authors used a Vav-cre system for the induction of ASXL2 deletion instead of Mx-cre. It would be interesting to compare the phenotype of these mice (HSPC number and some mature populations) to see if a chronic deletion of ASXL2 changes the phenotype observed.

4- Figure 1: in g, it is stated that the mice have a shorter lifespan, but there is no data on what happened to the mice. It would improve the manuscript to provide a description of the phenotype associated with the earliest death. Is it associated with a hematopoietic defect (splenomegaly, increased WBC in the peripheral blood)?

5- Figures 1 and 2: There is no data after 16 weeks post-transplantation (in the primary recipients). In the next part of the paper the authors suggest that ASXL2 loss is associated with an increased AML1-ETO leukemogenesis, it would be interesting to have data with later time points, to see if there is a long term effect of ASXL2 deficiency which could be associated with a pre-leukemic state.

6- Figure 4: It would reinforce the data found in this figure if the changes observed could be linked to gene expression data, to see if changes in histone modifications in enhancers are correlated with a change in gene expression. Moreover, it would be useful to provide global H3K27ac data as the change in H3K27ac in the ASXL2 KO cells is not convincing in this view of the browser.

7- The authors should include the list of genes with the Chip-seq data.

8- It would enhance the overall message of the manuscript to demonstrate that ASXL2 is recruited to the AML1-ETO target genes in the context of normal hematopoiesis, without AML1-ETO expression.

Minor comments:

- Page 2: need reference for ASXL1 expression –for example Fisher et al, 2006, Gene

- Page 8: "ASXL2 mutations are present in the predominant leukemic clone" – ref 11. It is not convincing that this reference shows this.

- Figure 1: (e-f).. as shown in (c): should be d

- Figure 5: (b) AML1-ETO9a or (c) AML1-ETO: it's the reverse in the figure

- Fig 1f and h: In f, it is shown that CD41+ Mk numbers decreased in ASXL2 KO BM, whereas in h the BM section has large number of (abnormal) Mks compared to WT; is this section representative of whole BM? Do remaining Mks just cluster together? Are these Mks CD41+?

- Figure 2d/Supp Figure 2a: In 2d the graph suggests a decrease in the number of HSCs, whereas in the FACS plot, the percentage is the same between WT and ASXL2 mice (0.010 vs 0.014, if we consider that the percentage represents the % in total population). Could the authors clarify this point?

- Figure 2: In the main text the figure 2 d makes reference to data obtained 16 weeks after transplantation, whereas in the figure legend it indicates that it is 8 weeks old mice without indication to transplantation. Could you please clarify this point?

- Page 3/supp Fig 1a-b: cycloheximide (n.b. spelling). The authors state mutant ASXL2 degrades more rapidly than WT, but only look at 1 timepoint (12 hrs). It should be stated that there is a greater loss of mutant than WT. The authors should either reword or perform a time course
- Supp fig 1c: are B6 the parental ES cells and 373 etc the targeted clones? If so, please state in legend.
- Supp table 1: The authors state that 2,986 genes are dysregulated in ASXL2 KO LSKs but 27,000 genes are listed in table – what is the cut-off for which genes are dysregulated?
- Supp Figure 1h: Are myeloid cells normal in ASXL2 KO mice? Neutrophils look hypogranular – are they dysplastic?

Reviewer #3 (Remarks to the Author): Expert in ATAC-seq

Summary

In this study Micol et al. demonstrate a functional role for ASXL2 mutations in the development of leukemia. Through the creation of a conditional knockout mouse the authors identified the requirement for ASXL2 in normal hematopoiesis. Combining ASXL2 mutations with the known AML1-ETO models identified a role for ASXL2 as a tumor suppressor. Overall this is a well-designed study demonstrating a novel role for ASXL2 in leukemia. There is some concern over the formatting of the manuscript and the interpretation of the ChIP-seq/ATAC-seq data that needs to be addressed. Generally, the authors should integrate their RNA-seq data from ASXL2 knockout cells with the ChIP-seq data to provide a mechanism for and improve the interpretation of the observed molecular changes. It is not clear if they are non-specific or due directly to loss of ASXL2.

Major Concerns:

1. This paper would benefit from section headers to guide the reader through the paper. Additionally, from what I can tell there is only a 1 paragraph discussion. The authors should expand this section to place their results in the context of the field.
2. The authors should expand on the ASXL2 ChIP-seq results since very little is presented and this is important information to present. Since the number of peaks seems low (324 total) the authors should present simple annotations of where the peaks are (TSS, intergenic, etc).
3. The comparison of ChIP-seq signal in Fig 3d does not indicate that ASXL2 binds fewer TSS since it is hard to compare the signal between different IPs. Also, are all TSS chosen or just a subset? The comparison is not accurate because there are likely different enrichment efficiencies between the antibodies and importantly there is a huge difference in the number of peaks between the samples. A comparison of the percentage of ASXL2 peaks that overlap TSS would be a better comparison.
4. Do the genes that are differentially expressed in ASXL2 mutants have ASXL2 binding sites by ChIP-seq? This would help classify the genes from direct and possibly indirect ASXL2 targets.
5. Since ASXL2 has so few binding sites how does it result in loss of H3K27me3 and increases in the active histone marks? Do these increases occur at ASXL2 binding sites? At TSS or enhancers for genes that are differentially expressed in ASXL2 mutant cells? Or non-specifically across the genome?
6. Similar to the above concerns, there is little description of the ATAC-seq changes observed. How many differences globally were observed? How do these changes correlate with observed histone/RNA-seq changes. Why were these loci chosen from the list of possible sites that changed?

Minor Concerns:

1. What is the scale for the CHIP-seq/ATAC-seq data and how were the data normalized? This should be added in the figure legend and described in the methods. i.e reads per million (rpm).
2. Where do the motifs in Supplementary Fig 2c rank in the entire list of motifs? Were these the top motifs or hand picked by the authors?

Reviewer #1: Expert in leukaemia and epigenetics

We thank the Reviewer for the kind compliments that our work provides " a mechanistic insight into the pathogenesis of AML M2 with ASXL2 mutations" and that the study is "well designed and the data are solid...with high impact uncovering for the first time the role of ASXL2 in hematopoiesis and leukemogenesis."

Major:

1) In Figure 3f, did the author compare with ASXL1 targets? Coexistence of ASXL2 with RUNX1 and AML-ETO is intriguing. Is there any physical interaction between ASXL2 and RUNX1 or AML-ETO? This is very important point. Can the author provide any molecular mechanisms for coexistence?

We thank the Reviewer and we agree that these are very important questions. To address the Reviewer's question regarding potential physical interaction of ASXL2 with RUNX1 and/or AML1-ETO we performed co-immunoprecipitation experiments where FLAG-tagged AML1-ETO cDNA or FLAG-tagged AML1-ETO9a cDNA were expressed followed by anti-FLAG immunoprecipitation and Western blotting for ASXL1 and ASXL2. These experiments failed to reveal any physical interaction between AML1-ETO or AML1-ETO9A and either of ASXL1 or ASXL2 despite efficient anti-FLAG pulldown. These data are now displayed in **Supplementary Figure 6a** and described in the Results and Discussion sections of the revised manuscript.

In addition to the above, we also performed co-immunoprecipitation experiments where FLAG-tagged ASXL1 or FLAG-tagged ASXL2 cDNA were expressed followed by anti-FLAG immunoprecipitation and Western blotting for RUNX1, SUZ12, BAP1, and FLAG. These experiments (1) failed to reveal any physical interaction between RUNX1 and either of ASXL1 or ASXL2 but (2) revealed previously described interaction between ASXL1 and SUZ12 (PUBMED ID 22897849), a core PRC2 member (PUBMED ID 14570930 and 15385962)). Of note, there was no clear interaction between ASXL2 and SUZ12, in contrast, suggesting that any interaction of ASXL family members with PRC2 in hematopoietic cells may be restricted to ASXL1. We also included Western blot analysis for BAP1 in these experiments as a positive control as BAP1 is known to interact with ASXL1 and ASXL2 based on prior reports (PUBMED ID 22878500). These data are now displayed in **Supplementary Figure 6b** and described in the Results and Discussion sections of the revised manuscript.

In Figure 3f we evaluated the overlap of ASXL2, RUNX1, and AML1-ETO gene targets due to the overlap in gene expression between *Asxl2* loss and RUNX1 and AML1-ETO direct binding sites (shown in Figure 3c). This overlap in ASXL2 dysregulated genes was further confirmed in new analyses shown in **Supplementary Figure 7b** revealing substantial overlap of RUNX1 and AML1-ETO binding sites with genes differentially expressed following ASXL2 loss in AML1-ETO expressing cells. Given that a similar overlap in genes dysregulated by *Asxl1* loss was not seen in Figure 3c combined with the lack of enrichment of ASXL1 mutations in t(8;21) AML (see PUBMED ID 27798625), we did not compare binding of ASXL1 to RUNX1 or AML1-ETO. However, we agree with the Reviewer that comparison of gene targets of ASXL1, ASXL2, RUNX1, and AML1-ETO will be interesting to evaluate in the future and this has been added to the revised Discussion.

2) Upon ASXL2 KD in SKNO-1 cells, how do the overlapping targets of ASXL2 and AML-ETO change in expression? What about in *Asxl2* KO HSPCs?

We thank the Reviewer for this insightful question. As mentioned above, to address this question we have now performed RNA-seq analysis of SKNO-1 cells treated with control shRNA or 2 different anti-ASXL2 shRNAs, each in biological triplicate. We used these data to analyze genes differentially expressed following ASXL2 loss in AML1-ETO leukemic cells and evaluate the overlap of differentially expressed genes with direct targets of ASXL2, RUNX1, or AML1-ETO. This revealed the following findings:

- We identified a number of genes known to promote leukemogenesis (either alone or in the context of AML1-ETO leukemia) as differentially expressed by ASXL2 loss. These include downregulation of *TET2* (see PUBMED IDs 25886910 and 26666262 for publications describing TET2

downregulation promoting AML1-ETO leukemogenesis) as well as *NOTCH2* (see PUBMED ID 21562564 which describes the role of NOTCH2 and NOTCH1 as tumor suppressors in myeloid leukemias) with ASXL2 loss in human *AML1-ETO* expressing cells. These data are now described in the Results section and **Supplementary Figure 7a** of the revised manuscript. In addition, the genes dysregulated by ASXL2 loss in these cells are listed in **Supplementary Tables 6-7**.

- We compared the genes differentially expressed following ASXL2 loss with those marked by RUNX1 or AML1-ETO direct binding in the same cell line. This revealed that ~90% of genes dysregulated in expression following ASXL2 loss are direct targets of RUNX1 or AML1-ETO. These data are described in the Results section and **Supplementary Figure 7b** of the revised manuscript.
- Regarding the overlap of AML1-ETO target genes with genes dysregulated following ASXL2 loss, the above data are consistent with the results shown in Figure 3c which reveal that a significant proportion of genes dysregulated in mouse hematopoietic stem/progenitor cells (HSPCs) following *Asxl2* deletion are also targets of RUNX1 and AML1-ETO.

Of note, the newly acquired RNA-seq data mentioned above have been added to the Gene Expression Omnibus (GEO) under accession number GSE84365 to ensure that these data are publicly available.

3) In Figure 4, did the H3K27me3 levels and enhancer histone marks change globally or specifically at the ASXL2 or AML-ETO targets? This point could be clarified by analyzing the histone changes at ASXL2/AML-ETO targets compared with non-targets.

We thank the Reviewer for these important questions. Based on the data in Figures 4a-b and further analyses as suggested by the Reviewer it appears that H3K27me3 levels changed across transcription start sites (TSSs) and gene bodies regardless of whether they were ASXL2 or AML1-ETO target genes. This was identified by analyzing H3K27me3 profiles at ASXL2 as well as AML1-ETO binding sites in SKNO-1 cells treated with control shRNA (shControl) or one of 2 different anti-ASXL2 shRNAs. As shown in **Supplementary Figures 7f-g**, H3K27me3 decreased at ASXL2 as well as at AML1-ETO binding sites. Although these changes in H3K27me3 were more prominent at ASXL2 binding sites than at 3 random control regions of the genome outside of ASXL2 peaks, based on the data in Figures 4a-b, H3K27me3 changes were prominent across TSS and gene bodies. These data have now been added to **Supplementary Figures 7f-g** and described in the revised Results section of the manuscript.

In addition to the above analyses, we also evaluated H3K4me1 levels with or without ASXL2 loss at ASXL2 binding sites. As shown in **Supplementary Figure 7h**, there was no clear correlation between changes in H3K4me1 and sites of ASXL2 binding.

Did H3K27me3 also change in primary *Asxl2* KO HSPCs? What is the mechanism underlying this phenomenon?

To address this question we have now performed Western blot analysis for H3K27me3, H3K27Ac, H3K4me1, and total histone H3 on purified histones from bone marrow mononuclear cells from *Asxl2*-deficient, *Asxl1*-deficient, or *Asxl1/2* wildtype bone marrow mononuclear cells (unfortunately it was not feasible to do Western blot analysis for histones using hematopoietic stem/progenitor cells given the cell numbers required for Western blot on purified histones). This revealed no global changes in these marks in *Asxl2* knockout bone marrow cells. In contrast, reduced H3K27me3 was evident in *Asxl1* knockout bone marrow cells. These data have been added to **Supplementary Figure 6f** and described in the Results section of the revised manuscript. In addition, we have also now performed anti-H3K27me3 ChIP-seq in c-Kit⁺ cells from *Mx1-cre* control, *Mx1-cre Asxl1^{fl/fl}*, and *Mx1-cre Asxl2^{fl/fl}* mice (cells were collected 4 weeks following plpC administration to 6-week old mice). Similar to the histone Western blotting results, these data also revealed a greater decrease in H3K27me3 in *Asxl1* deficient mice over other groups (and at this time point loss of H3K27me3 was not evident in *Asxl2* mice). These data have been added to **Supplementary Figure 6g** and described in the Results section of the revised manuscript. In addition, the additional ChIP-seq data have been uploaded to the GEO database under accession number GSE84365.

Does ASXL2 interact with PRC2 as ASXL1 does?

We thank the Reviewer for this important question. As noted above, we performed co-immunoprecipitation experiments where FLAG-tagged ASXL1 or FLAG-tagged ASXL2 cDNA were expressed followed by anti-FLAG immunoprecipitation and Western blotting for RUNX1, SUZ12, BAP1, and FLAG. These experiments failed to reveal any physical interaction between RUNX1 and either of ASXL1 or ASXL2 but revealed a previously described interaction between ASXL1 and the core PRC2 member SUZ12. Again, Western blotting for BAP1 was included in these experiments as a positive control as it is known to interact with ASXL1 and ASXL2 based on prior reports (PUBMED ID 22878500). These data are now displayed in **Supplementary Figure 6b** and described in the Results and Discussion sections of the revised manuscript.

4) Please show the global changes in H3K27me3, H3K4me1, H3K27ac in *Asxl2* KO HSPCs by Western blot analysis.

We thank the Reviewer for this important question. As mentioned above, we have now performed Western blot analysis for H3K27me3, H3K4me1, H3K27Ac, and total histone H3 on purified histones from *Asxl1*-deficient, *Asxl2*-deficient, or control bone marrow mononuclear cells. This revealed no global changes in these marks in *Asxl2* knockout bone marrow cells. In contrast, reduced H3K27me3 was evident in *Asxl1* knockout bone marrow cells. These data have been added to **Supplementary Figure 6f** and described in the Results sections of the revised manuscript.

5) Please show the correlation between histone changes (H3K27me3 at promoters and enhancer marks) and gene expression. Is there clear correlation?

We thank the Reviewer for this question. We have now performed RNA-seq analysis of SKNO-1 cells treated with control shRNA (shControl) or one of two different anti-ASXL2 shRNAs in biological triplicate (these data are now shown in **Supplementary Figure 7a** and **Supplementary Tables 6-7**). To address the Reviewer's question we then evaluated the effect of ASXL2 loss on the expression of genes whose promoters are marked by H3K27me3 in shControl cells using Gene Set Enrichment Analysis (GSEA). This revealed that genes marked by H3K27me3 in control cells experienced upregulation upon ASXL2 knockdown (although this result was not statistically significant). These results have been added to **Supplementary Figure 7c**.

In addition to the above, we also attempted to evaluate the effect of ASXL2 loss on the expression of genes within 100kB of H3K4me1 or H3K27Ac peaks in shControl cells. This analysis failed to reveal a clear effect on gene expression as shown in **Supplementary Figure 7d-e**. This may be explained by the fact that these are putative enhancers that do not necessarily regulate the expression of genes located in 100kB proximity of these genes. These data have been added to **Supplementary Figure 7d-e**.

Minor:

1. Do *Asxl2* KO show dysplasia of myeloid cells? Are the hematopoietic defects observed in *Asxl2* KO compatible with MDS? What is the cause of early death of the mice?

Based on the Reviewer's question and salient observations from Reviewer #2, we have now carefully re-evaluated the histomorphology of bone marrow, spleen, and peripheral blood of *Mx1-cre Asxl2^{fl/fl}* mice. This has revealed the following:

- 1) There is evidence of hyposegmented neutrophils with hypogranular cytoplasm and circulating, multinucleated erythroid progenitors, which are features consistent with myelodysplasia. These data have now been added to a new **Supplementary Figure 2**.
- 2) Per comments from Reviewer #2, we have confirmed the clustering of megakaryocytes with overall reduced megakaryocytes in the bone marrow of *Asxl2* null mice. New images revealing these findings have been added to a new **Supplementary Figure 2**.

Regarding the cause of early death of mice with hematopoietic-specific deletion of *Asxl2*, we apologize

that these data were not described in the original manuscript submission. We have never observed any transplanted mice or primary *Asxl2* knockout mice that had any evidence of elevated WBC counts or splenomegaly at any time point. We have analyzed a cohort of *Mx1-cre Asxl2^{fl/fl}* and *Mx1-cre* control transplanted mice up to 52 weeks (data shown in a new **Supplementary Figure 3**). During this period of observation we did not observe any *Asxl2* homozygous knockout mice that had evidence of elevated WBC count or splenomegaly. Moreover, even at these time points, *Asxl2* loss was associated with decreased white blood cell and platelet counts and at 52 weeks post-transplant, *Asxl2* knockout mice (*Mx1-cre Asxl2^{fl/fl}*) still exhibited reduced number of hematopoietic stem and progenitor cells relative to littermate, age-matched *Mx1-cre* control mice. Overall these data suggest that *Asxl2*-deficient hematopoietic cells are associated with functional defects in hematopoiesis that are associated with shortened survival in the mice. These data have been added to a new **Supplemental Figure 3** and described in the revised Results and Discussion sections.

2. In Figure 5c, AML1-ETO9a is highly potent. Is there any functional relationship between AML1-ETO9a and ASXL2? Does AML1-ETO9a antagonize ASXL2 function or recruitment to its targets? If anything known, please discuss?

We thank the Reviewer for this interesting question. The potency of AML1-ETO9a, an alternatively spliced transcript of *AML1-ETO*, relative to AML1-ETO in leukemogenesis is previously described (see PUBMED ID 16892037) and was evident in *Asxl2* wildtype as well as *Asxl2*-deficient backgrounds. It is therefore quite possible that the potency of AML1-ETO9a in Figure 5c is simply a reflection of this previously described enhanced potency of AML1-ETO9a compared with AML1-ETO. However, this does not exclude a specific interplay of AML1-ETO9a and ASXL2. To this end, we did not detect physical interaction between AML1-ETO9a and ASXL2 (as described above and now shown in **Supplementary Figure 6a**). We have therefore clarified in the Discussion that the increased potency of AML1-ETO9a in an *ASXL2* mutant background may simply indicate the known increased leukemogenicity of AML1-ETO9a relative to AML1-ETO and/or suggest a potential genetic interaction between *ASXL2* loss and AML1-ETO9a, which will be interesting to study in the future.

3. Does dysfunction of ASXL1 also collaborate with AML-ETO in leukemogenesis? Please discuss.

We thank the Reviewer for this question. While mutations in *ASXL1* have been described in patients with t(8;21) AML, one very recent large whole exome/whole genome sequencing study of t(8;21) AML (PUBMED ID 27798625) did not find a single *ASXL1* mutation in a cohort of 85 t(8;21) AML adult and pediatric patients. This is in stark contrast to the 19.8% frequency of *ASXL2* mutations in t(8;21) AML noted in this same study (PUBMED ID 27798625). Overall, across the 2 largest sequencing studies of t(8;21) AML where both *ASXL1* and *ASXL2* have been sequenced it appears that *ASXL2* mutations are present in 21% of patients (39/187) in contrast to the 5.9% of *ASXL1* mutations (11/187) in t(8;21) AML (PUBMED IDs 27798625 and 26980726). These data, combined with the functional data in this manuscript identifying transcriptional overlap of *RUNX1* and *AML1-ETO* target genes following *Asxl2* loss but not *Asxl1* loss, suggest that *ASXL1* mutations and AML1-ETO may not collaborate in the manner seen here with *ASXL2* loss and AML1-ETO. This will be an important question for future studies attempting to decipher functional differences between *ASXL1* and *ASXL2* further and we have added these thoughts and directions for future efforts to the revised Discussion.

Reviewer #2: Expert in haematopoiesis

We thank the Reviewer for the kind complements that "the experiments were well performed and the results are interesting."

Major comments:

1) More data on the primary Mx-cre ASXL2 model, without transplantation, should be provided, especially on the numbers of the different bone marrow progenitor populations (HSC and GMP in particular) and mature populations.

We thank the Reviewer for this point and we have now added data on the phenotypic effects of heterozygous as well as homozygous deletion of *Asx12* in primary mice without transplantation using *Mx1-cre* control, *Mx1-cre Asx12fl/wt*, and *Mx1-cre Asx12fl/fl* mice. These data have now been added to **Supplementary Figure 4b-f** and reveal the following findings:

- Analysis of longitudinal blood counts up to 26 weeks post-plpC administration revealed that loss of either 1 or 2 copies of *Asx12* is consistently associated with reduced total peripheral white blood cell counts (primarily due to reduced B220⁺ cells) and platelet counts compared with *Mx1-cre Asx12* wildtype littermate controls treated with plpC. These results are shown in **Supplementary Figure 4b-c** and are analogous to results with noncompetitive transplantation shown in Figure 1.
- Analysis of the effects of *Asx12* deletion on hematopoietic stem and progenitor cells in *Mx1-cre* control, *Mx1-cre Asx12fl/wt*, and *Mx1-cre Asx12fl/fl* mice at 26 weeks post-plpC administration. These data revealed significant reduction in absolute numbers of multipotent progenitors (MPPs), LSK cells, and megakaryocyte/erythroid progenitor (MEP) cells as well as percentage of splenic erythroid precursors (CD71⁺ Ter119⁻ cells) and megakaryocytes (CD41⁺ cells) in *Mx1-cre Asx12fl/fl* mice compared with *Mx1-cre* control mice. *Mx1-cre Asx12fl/wt* mice also exhibited significant reductions in absolute numbers of MEPs as well as splenic erythroid precursors (CD71⁺ Ter119⁻ cells) and megakaryocytes (CD41⁺ cells) compared with *Mx1-cre* control mice. These data are all now shown in **Supplementary Figure 4d-f** and described in the Results section of the revised manuscript.

2) Page 6: "ASXL2 is haploinsufficient with regards to normal HSPC function and number" – the authors show that HSPC function is affected but it is unclear if they can say numbers are reduced as they don't show data for non-transplanted transgenic ASXL2 het mice

We completely agree with the Reviewer on this point and have now edited this statement to remove the word "number."

3) In the figure 5, the authors used a *Vav-cre* system for the induction of ASXL2 deletion instead of *Mx-cre*. It would be interesting to compare the phenotype of these mice (HSPC number and some mature populations) to see if a chronic deletion of ASXL2 changes the phenotype observed.

We agree with the Reviewer that detailed phenotypic analysis of *Vav-cre Asx12fl/fl* and *Vav-cre Asx12fl/wt* mice would be interesting to determine the effect of chronic deletion of *Asx12*. However, the purpose of this experiment utilizing *Vav-cre* mice was to evaluate cooperativity between *Asx12* loss and AML1-ETO expression using an additional system beyond the *Mx1-cre* system. While evaluation of the effects of *Asx12* deletion in the hematopoietic system using *Vav-cre* will be useful, unfortunately we do not have enough *Vav-cre Asx12fl/fl* mice to evaluate the phenotypes of these animals. We sincerely apologize for this and hope that the generation of *Asx12* conditional knockout mice and numerous compound *Asx12* knockout mice here for the first time (which include *Mx1-cre Asx12fl/fl Asx12fl/fl* mice, *Mx1-cre Asx12fl/fl Aml1-Eto* conditional knockin mice, *Mx1-cre Asx12fl/fl* with retroviral overexpression of AML1-ETO9A, *Mx1-cre Asx12fl/fl Aml1-Eto* conditional knockin mice with retroviral overexpression of NRASG12D) would be sufficient for the initial manuscript describing this model. On a related note, our manuscript was co-submitted with a manuscript by Dr. Feng-Chun Yang who has studied the effects of *Asx12* loss on hematopoiesis using a completely separate mouse model. In light of this, we hope the

reviewer will agree that the presentation across both studies will provide data on the role of *Asxl2* in hematopoiesis using multiple mouse models already. Finally, we are also in the process of publically depositing *Asxl2* floxed mice at Jackson laboratories so that researchers may study these mice to address questions such as this without constraints.

4) Figure 1: in g, it is stated that the mice have a shorter lifespan, but there is no data on what happened to the mice. It would improve the manuscript to provide a description of the phenotype associated with the earliest death. Is it associated with a hematopoietic defect (splenomegaly, increased WBC in the peripheral blood)?

We thank the Reviewer for this important question and apologize that these data were not described in the original manuscript submission. We have never observed any transplanted mice or primary *Asxl2* knockout mice to have elevated WBC counts or splenomegaly at any time point. This includes evaluation of a cohort of *Mx1-cre Asxl2^{fl/fl}* and *Mx1-cre* control transplanted mice up to 52 weeks post-transplantation (a time point in which the earliest deaths of *Mx1-cre Asxl2^{fl/fl}* mice were captured). During this period of observation we did not observe any *Asxl2* homozygous knockout with an elevated WBC count or splenomegaly. Moreover, even at these time points, *Asxl2* loss was associated with decreased WBC and platelet counts and at 52 weeks post-transplant *Asxl2* knockout mice (*Mx1-cre Asxl2^{fl/fl}*) still exhibited reduced number of hematopoietic stem and progenitor cells relative to littermate, age-matched *Mx1-cre* control mice. These data are shown in a new **Supplemental Figure 3** and described in the Results section of the revised manuscript.

In addition, we analyzed a cohort of primary *Mx1-cre* control, *Mx1-cre Asxl2^{fl/WT}*, and *Mx1-cre Asxl2^{fl/fl}* mice at 26 weeks of age and at these ages the *Asxl2* deficient mice exhibited no evidence of elevated WBC count or splenomegaly. Overall these data suggest that *Asxl2*-deficient hematopoietic cells are associated with functional defects in hematopoiesis that are associated with shortened survival in the mice and we apologize that we are unable to clarify this issue with any greater detail. The primary mouse data have now been added to **Supplemental Figure 4b-f** and we have described these data in the revised Results section. In addition, we have described this in the Discussion section of the revised manuscript now as well.

5) Figures 1 and 2: There is no data after 16 weeks post-transplantation (in the primary recipients). In the next part of the paper the authors suggest that *ASXL2* loss is associated with an increased AML1-ETO leukemogenesis, it would be interesting to have data with later time points, to see if there is a long term effect of *ASXL2* deficiency which could be associated with a pre-leukemic state.

We thank the Reviewer for this question. In response to this point, we have now added a new **Supplemental Figure 3** which shows phenotypic evaluation of non-competitively transplanted *Mx1-cre Asxl2^{fl/fl}* mice and *Mx1-cre Asxl2* wildtype mice up to 52 weeks post-transplantation. Regarding the question of a potential pre-leukemic state with *Asxl2* loss, as noted above, despite following primary and transplanted *Asxl2* deficient mice for 26 and 52 weeks respectively, we did not identify any mice with *Asxl2* loss alone to have developed increased white blood counts, splenomegaly, or expansion of hematopoietic stem/progenitor cells. These data have now been added to **Supplemental Figures 3 and 4** and described in the revised Results and Discussion sections.

Overall, these data suggest that loss of *ASXL2* is not compatible with normal hematopoiesis on its own but promotes leukemogenesis in the setting of AML1-ETO expression. These are findings that are consistent with human genetic data where *ASXL2* is not known to be mutated in the setting of clonal hematopoiesis (in stark contrast with *ASXL1* which is commonly mutated in pre-leukemic settings (PUBMED IDs 25426838, 25426837, and 25732814)). In contrast, *ASXL2* is frequently mutated in AML1-ETO leukemia specifically. These genetic data along with our functional evaluation of the effects of *ASXL2* loss alone and in the context of AML1-ETO expression, suggest that *ASXL2* loss contributes to a clonal advantage in the context of the AML1-ETO fusion oncoprotein but not in otherwise normal hematopoietic stem cells without expression of AML1-ETO. We have now clarified these points in a much-expanded Discussion section of the revised manuscript.

6) Figure 4: It would reinforce the data found in this figure if the changes observed could be linked to gene expression data, to see if changes in histone modifications in enhancers are correlated with a change in gene expression. Moreover, it would be useful to provide global H3K27ac data as the change in H3K27ac in the ASXL2 KO cells is not convincing in this view of the browser.

We thank the Reviewer for these suggestions, some of which were also echoed by Reviewers #1 and #3. To address these questions we have now performed RNA-seq of SKNO-1 cells treated with control shRNA or 1 of 2 different anti-ASXL2 shRNAs. We then utilized these data to correlate alterations in gene expression with ASXL2 loss at genes marked by H3K27me3 at promoters or within 100kB of H3K4me1 or H3K27Ac peaks in shControl cells. These analyses revealed:

- Genes whose promoters were marked by H3K27me3 in control cells experienced upregulation upon ASXL2 knockdown (although this result was not statistically significant).
- Genes within 100kB of H3K4me1 or H3K27Ac peaks in shControl cells exhibited no consistent alteration in gene expression with ASXL2 loss. The lack of correlation of alteration in gene expression with H3K4me1 abundance may be explained by the fact that these are putative enhancers that do not necessarily regulate the expression of genes located in 100kB proximity of these genes.

All of the above data are explained in the revised Results sections of the manuscript and have been added to **Supplementary Figure 7c-e**.

Of note, the analysis of differentially expressed genes following ASXL2 loss in SKNO-1 cells is now included in a new **Supplemental Figure 7a** and lists of differentially expressed genes are located in **Supplementary Tables 6-7**. Finally, all newly generated RNA-seq data have been uploaded to the GEO database (under accession number GSE84365).

We have also performed Western blot analysis of bone marrow mononuclear cells extracted from *Mx1-cre Asxl2^{fl/fl}* mice for a number of histone modifications, including H3K27Ac. This did not reveal global alterations in H3K27Ac with *Asxl2* loss in mice (data are shown in a new **Supplemental Figure 6f**). Please note this is not an unexpected result as the data in the manuscript reveal changes in *putative enhancers simultaneously marked by H3K4me1 and H3K27Ac*. We did not attempt to give the impression that H3K27Ac is globally changed with ASXL2 loss or even that H3K27Ac alone is changed at enhancers. We have now edited the manuscript to clarify this point and apologize if this was not clear in the prior version of the manuscript.

7) The authors should include the list of genes with the ChIP-seq data.

We thank the Reviewer for pointing this out and we apologize for the oversight of not including this data in the initial submission. We have now included the list of genes associated with anti-ASXL2, RUNX1, and AML1-ETO ChIP-seq to **Supplemental Tables 2-4**, respectively.

8) It would enhance the overall message of the manuscript to demonstrate that ASXL2 is recruited to the AML1-ETO target genes in the context of normal hematopoiesis, without AML1-ETO expression. We thank the Reviewer for this point. In the revised manuscript we demonstrate an overlap in genes dysregulated by ASXL2 loss and targets of RUNX1 and AML1-ETO in multiple cell types (please see Figure 3c and the new **Supplementary Figure 7b**). We therefore have attempted to directly demonstrate that ASXL2 is recruited to AML1-ETO target genes in normal hematopoietic cells but unfortunately despite numerous attempts at anti-*Asxl2* ChIP-seq in c-Kit⁺ cells from wildtype C57/Bl6 mice we were unsuccessful at generating ChIP-seq data with reasonable quality (this includes trials with multiple commercial anti-ASXL2 antibodies). We have mentioned the importance of demonstrating this in future work in the Discussion and are currently generating bespoke anti-*Asxl2* antibodies and other methods to attempt to address the need for higher quality anti-*Asxl2* ChIP-seq data.

Minor comments:

- Page 2: need reference for ASXL1 expression –for example Fisher et al, 2006, Gene
We apologize to the Reviewer for this oversight and we have now added this reference to the manuscript.

- Page 8: "ASXL2 mutations are present in the predominant leukemic clone" – ref 11. It is not convincing that this reference shows this.

We thank the Reviewer for pointing this out and have now rephrased this sentence to more precisely state that "the average variant allele frequency of mutations in ASXL2 are significantly higher than those of *c-KIT*, *FLT3*, or *N/KRAS*." This data is shown in Supplementary Figure 4 of PUBMED ID 26980726 and we have included the relevant parts of this previously published data with more detailed information below. As shown below, ASXL2 mutations have substantially higher variant allele frequency (VAF) than mutations in FLT3, RAS, and RAS kinases in patients with t(8;21) AML:

- Figure 1: (e-f) as shown in (c): should be d

We apologize to the Reviewer for this oversight and we have now made this correction in the Figure legend text.

- Figure 5: (b) AML1-ETO9a or (c) AML1-ETO: it's the reverse in the figure

We apologize to the Reviewer for this error and we have now made this correction in the Figure legend text.

- Fig 1f and h: In f, it is shown that CD41+ Mk numbers decreased in ASXL2 KO BM, whereas in h the BM section has large number of (abnormal) Mks compared to WT; is this section representative of whole BM? Do remaining Mks just cluster together? Are these Mks CD41+?

We thank the Reviewer for this astute observation. We have now re-evaluated bone marrow (BM) sections from the mice analyzed in the prior Supplementary Figure 1h. The Reviewer is indeed correct that the BM section shown for *Mx1-cre Asxl2fl/fl* mice in the prior Supplementary Figure 1h is not representative of the entire BM of *Mx1-cre Asxl2fl/fl* mice but was chosen to highlight megakaryocyte clustering. To make this point more clear, we now show multiple sections of *Mx1-cre Asxl2fl/fl* BM in a new **Supplementary Figure 2**. In addition, we have performed anti-CD41 immunohistochemistry and H&E staining on corresponding sections of BM, which confirm that the megakaryocytes in all 4 genotypes of mice (*Mx1-cre* control, *Mx1-cre Asxl1fl/fl*, *Mx1-cre Asxl2fl/fl*, and *Mx1-cre Asxl1fl/fl Asxl2fl/fl*) express CD41 (shown below):

Scale Bar = 100 μ m; 600x magnification

- Figure 2d/Supp Figure 2a: In 2d the graph suggests a decrease in the number of HSCs, whereas in the FACS plot, the percentage is the same between WT and ASXL2 mice (0.010 vs 0.014, if we consider that the percentage represents the % in total population). Could the authors clarify this point? We apologize to the Reviewer for the lack of clarity on this point regarding total numbers and percentages of hematopoietic stem and progenitor cell populations. **Figure 2d** shows the number of each cell population as an average across 10 different mice per group. In contrast, the prior Supplementary Figure 2a (now **Supplementary Figure 4a**) shows the percentages of each population (from the parent gate and not any other larger population) from a single representative mouse. The parent gate is listed above each FACS plot in **Supplementary Figure 4a** and for HSCs the parent gate is LSK cells which are reduced in percentage and number in the representative mice shown in this figure.

- Figure 2: In the main text the figure 2 d makes reference to data obtained 16 weeks after transplantation, whereas in the figure legend it indicates that it is 8 weeks old mice without indication to transplantation. Could you please clarify this point? We apologize to the Reviewer for the confusion on Figure 2d. These data are indeed from transplanted mice following 16-weeks of plpC and this point has now been clarified in the figure legend in the revised manuscript.

- Page 3/supp Fig 1a-b: cycloheximide (n.b. spelling). The authors state mutant ASXL2 degrades more rapidly than WT, but only look at 1 timepoint (12 hrs). It should be stated that there is a greater loss of mutant than WT. The authors should either reword or perform a time course. We thank the Reviewer for this suggestion and we have revised the text as suggested to state there is "greater loss of mutant than WT" as suggested by the Reviewer. We also corrected the spelling of cycloheximide in the revised manuscript and thank the Reviewer for noticing this error.

- Supp fig 1c: are B6 the parental ES cells and 373 etc the targeted clones? If so, please state in legend.

The Reviewer is correct regarding the meaning of the labeling in Supplementary Figure 1c and we have now revised the figure legend accordingly. We apologize for this oversight.

- Supp table 1: The authors state that 2,986 genes are dysregulated in ASXL2 KO LSKs but 27,000 genes are listed in table – what is the cut-off for which genes are dysregulated?
We apologize to the Reviewer that the criteria used for determining statistical significance was not clear in the initial submission of the manuscript. For the gene expression data shown in **Figures 3a-b** and **Supplementary Fig. 7a**, we used a criterion of *p-value adjusted for multiple comparisons* of < 0.05 to determine significance. This is now described in the Figure legend for each figure. **Supplementary Table 1** and the new **Supplementary Tables 6-7** instead list all differentially expressed genes based on fold-change and uncorrected *p*-value to provide readers who may be interested as comprehensive of a list of differentially expressed genes as possible.

- Supp Figure 1h: Are myeloid cells normal in ASXL2 KO mice? Neutrophils look hypogranular – are they dysplastic?
We thank the Reviewer for this astute observation and a similar question was asked by Reviewer #1. Based on the Reviewers' questions, we have now carefully re-evaluated the histomorphology of bone marrow, spleen, and peripheral blood of *Mx1-cre Asxl2^{fl/fl}* mice. This has revealed evidence of hyposegmented neutrophils with hypogranular cytoplasm and circulating, multinucleated erythroid progenitors, which are features consistent with myelodysplasia. These data have now been added to a new **Supplementary Figure 2**.

Reviewer #3: Expert in ATAC-seq

We thank the Reviewer for the complement that this "this is a well-designed study demonstrating a novel role for ASXL2 in leukemia."

Major Concerns:

1. This paper would benefit from section headers to guide the reader through the paper. Additionally, from what I can tell there is only a 1 paragraph discussion. The authors should expand this section to place their results in the context of the field.

We thank the Reviewer for this suggestion and we completely agree that section headers and an expanded Discussion would improve the readability of the paper (the originally submitted manuscript format was a result of this manuscript being formatted for another Nature publishing group journal before being transferred internally to *Nature Communications*). We have now divided the manuscript into Introduction, Results, and Discussion sections and subdivided the Results section with section headers. We have also substantially expanded the Discussion (it now consists of 5 paragraphs and is approximately 5 times the length of the prior version of the Discussion).

2. The authors should expand on the ASXL2 ChIP-seq results since very little is presented and this is important information to present. Since the number of peaks seems low (324 total) the authors should present simple annotations of where the peaks are (TSS, intergenic, etc).

We thank the Reviewer for this important point and we have now added further analysis and description of the anti-ASXL2 ChIP-seq data in SKNO-1 cells. These data now include annotation of ASXL2 binding peaks throughout genomic regions (including intergenic regions, introns, promoters, 3' UTRs, 5' UTRs, exons, non-coding RNAs, promoters, and transcription termination sites) as well as histograms of anti-ASXL2 ChIP-seq reads at gene bodies as well as transcription start sites (TSS). These data have been added to **Supplementary Figure 5b-c** and are described in the revised Results section of the manuscript.

3. The comparison of ChIP-seq signal in Fig 3d does not indicate that ASXL2 binds fewer TSS since it is hard to compare the signal between different IPs. Also, are all TSS chosen or just a subset? The comparison is not accurate because there are likely different enrichment efficiencies between the antibodies and importantly there is a huge difference in the number of peaks between the samples. A comparison of the percentage of ASXL2 peaks that overlap TSS would be a better comparison.

We thank the Reviewer for these valid points. In Figure 3d, all TSS were considered. We have now clarified this point in the revised manuscript and mentioned the caveats in the Results section of (i) likely different enrichment efficiencies between antibodies and (ii) different number of peaks across the different immunoprecipitated proteins. Given these issues, we have now also evaluated the percentage of ASXL2, AML1-ETO, and RUNX1 peaks that overlap with promoters (defined as -2.5 Kb to +2.5 Kb of TSS). The percentage of TSS-overlapping peaks for ASXL2, AE and RUNX1 are 9.7%, 27%, and 29.8% respectively. These data suggest that ASXL2 binds fewer TSS than AE and RUNX1 and these results have been added to the Results section immediately following clarification of caveats of comparison of number of peaks between samples.

4. Do the genes that are differentially expressed in ASXL2 mutants have ASXL2 binding sites by ChIP-seq? This would help classify the genes from direct and possibly indirect ASXL2 targets.

We thank the Reviewer for this astute point. In order to address this question, we have now performed RNA-seq of SKNO-1 cells with or without shRNA knockdown of ASXL2. We then evaluated how many of the genes up- or down-regulated following ASXL2 loss are bound by ASXL2, RUNX1, and/or AML1-ETO. These data revealed that >90% of ASXL2 dysregulated genes are RUNX1 or AML1-ETO target genes while a far smaller portion of ASXL2 dysregulated genes are actually bound by ASXL2. This suggests that a number of genes dysregulated following ASXL2 loss are not direct targets of ASXL2—a point that is not totally unexpected given the changes in H3K27me3 and active and poised enhancers

following ASXL2 loss. These data have now been added to the manuscript in **Supplementary Figure 7** and described in the Results and Discussion sections of the revised manuscript.

5. Since ASXL2 has so few binding sites how does it result in loss of H3K27me3 and increases in the active histone marks? Do these increases occur at ASXL2 binding sites? At TSS or enhancers for genes that are differentially expressed in ASXL2 mutant cells? Or non-specifically across the genome? We thank the Reviewer for these important questions (some of which were mirrored by Reviewers #1 and #2). Based on the data in Figures 4a-b and further analyses as suggested by the Reviewer, it appears that H3K27me3 levels changed across transcription start sites (TSSs) and gene bodies regardless of whether they were ASXL2 or AML1-ETO binding sites. This was identified by analyzing H3K27me3 ChIP-seq profiles at ASXL2 as well as AML1-ETO binding sites in SKNO-1 cells treated with control shRNA (shControl) or one of 2 different anti-ASXL2 shRNAs. As evidenced from **Supplementary Figures 7f-g**, H3K27me3 decreased at ASXL2 as well as AML1-ETO binding sites. Although these changes in H3K27me3 were more prominent at ASXL2 binding sites than at random control regions of the genome outside of ASXL2 peaks, based on the data in Figures 4a-b, H3K27me3 changes were prominent across TSS and gene bodies. These data have now been added to **Supplementary Figures 7f-g** and described in the Results section of the revised manuscript.

In addition to the above analyses, we also evaluated H3K4me1 levels with or without ASXL2 loss at ASXL2 binding sites. As shown in **Supplementary Figure 7h**, there was no clear correlation between changes in H3K4me1 and sites of ASXL2 binding.

Based on the above data, the change in H3K27me3 and enhancers seen with ASXL2 loss is unlikely to be attributable to direct binding of ASXL2 to the genome in addition to the modest number of ASXL2 binding sites (as noted by the Reviewer). We therefore believe that the effects of ASXL2 loss on chromatin state are therefore likely to be indirect effects. These points have all been added to the revised manuscript in the Results and Discussion sections. Although the precise molecular mechanism by which ASXL2 loss is associated with these changes in chromatin state is not yet clear, this is a very important avenue of ongoing and future research for our group and others. We have described the need to understand the molecular mechanisms by which ASXL2 loss is associated with these changes in chromatin state in the revised Discussion in addition to possible explanations for the relationship between ASXL2 loss and alterations in enhancers.

6. Similar to the above concerns, there is little description of the ATAC-seq changes observed. How many differences globally were observed? How do these changes correlate with observed histone/RNA-seq changes. Why were these loci chosen from the list of possible sites that changed? We thank the Reviewer for this important suggestion to add more detailed description of global ATAC-seq results to the manuscript. We have now added these results to a new **Supplementary Figure 10** which show that global chromatin accessibility as assessed by ATAC-seq is not substantially different in AML1-ETO9a/*Mx1*-cre *Asx/2* wildtype versus AML1-ETO9a/*Mx1*-cre *Asx/2fl/fl* leukemias. However, as shown in **Figure 5h** there was a substantial and consistent increase in chromatin accessibility at the *HoxA* and *Meis1* loci AML1-ETO9a/*Mx1*-cre *Asx/2fl/fl* leukemia samples relative to AML1-ETO9a/*Mx1*-cre *Asx/2* wildtype samples. We highlighted the ATAC-seq changes at the *HoxA* and *Meis1* cluster for two reasons which are now more fully explained in the Results and Discussion sections of the revised manuscript. First, *Asx* proteins were originally identified as being required for assisting Polycomb and Trithorax group proteins in regulating chromatin state and expression of *Hox* genes (see PUBMED IDs 9477319 and 10394912). Second, upregulation of *HoxA* genes is a well-established mechanism of leukemogenesis (see PUBMED ID's 25600023, 11113197, 8972230, and 20130239 as a few examples of studies highlighting the leukemogenicity of upregulation of nearly any *HoxA* gene cluster member). These points have now been highlighted in the Results and Discussion section and we apologize that a more detailed discussion of the context of these findings were lacking in the prior version of the manuscript.

Minor Concerns:

1. What is the scale for the ChIP-seq/ATAC-seq data and how were the data normalized? This should be added in the figure legend and described in the methods. i.e reads per million (rpm).

We apologize to the Reviewer for the lack of clarity on this issue. The scale for ChIP-seq as well as ATAC-seq used throughout the manuscript is read counts per million mapped reads. We have now ensured that this information is included in the legends for every main and Supplemental figure where ChIP- or ATAC-seq data are shown. We have also noted in the revised Methods section that ChIP- and ATAC-seq data were normalized by evaluating read counts per million mapped reads.

2. Where do the motifs in Supplementary Fig 2c rank in the entire list of motifs? Were these the top motifs or hand picked by the authors?

We thank the Reviewer for this question, which we have now clarified in the revised manuscript. The motifs shown in the prior Supplementary Fig. 2c (now currently **Supplementary Fig. 5d**) were handpicked among the ~66 significantly enriched motifs in the anti-ASXL2 ChIP-seq data due to their being amongst the most enriched motifs and the known biological importance of these motifs to RUNX1 and/or AML1-ETO binding. We have now explained this point further in the revised Results section of the manuscript and also added a new **Supplementary Table 5** with all of the significantly enriched motifs identified from the anti-ASXL2 ChIP-seq data.

REVIEWERS' COMMENTS:

Reviewer #1 (Remarks to the Author):

The authors have done enough experiments for revision and responded to most of my concerns precisely. The manuscript is now acceptable for publication.

Reviewer #2 (Remarks to the Author):

The authors have performed thorough revisions and addressed my concerns. I just would like to offer some minor comments/queries to their rebuttal:

- a. Major comment 6/Supplementary Figure 7c: The authors have examined the correlation between H3K27Me3 marks and differential gene expression upon ASXL2 knockdown. However, given the P values of 0.4 and 0.8, I am not sure they can state that there is a correlation.
- b. Major comment 6/Figure 4f: It is difficult to visualize where on the HOXA gene locus the H3K27Ac peaks are altered. Could the authors perhaps highlight them by putting boxes around significantly altered peaks?
- c. Minor comment relating to Figure 2d/Supplementary Figure 4a: There still seems to be some confusion about the percentages displayed in the FACS plots in the bottom half of Supplementary Figure 4a ("Gated on LSK from the above"); the authors state that the figure shows the percentages of each population from the parent gate. In the bottom figures, the numbers are not the percentages of the parent "LSK" gate but of the grandparent "live, CD45/2+, lineage-negative cells" gate. Please could the authors amend this, as the CD150+ CD48- HSC fraction appears increased as a proportion of the LSK in the Asxl2 and Asxl1/Asxl2 knockout mice at the expense of the CD48+ CD150- fraction and this should be made clearer.

Reviewer #3 (Remarks to the Author):

The authors present a well revised and improved manuscript and in general have addressed all of my concerns. However, there are two remaining points that should be addressed.

1. In regards to the overlap of Asxl2 signal at TSS presented in Figure 3d, the histogram presented should be interpreted as enrichment and not binding. In the text on pg 9 the sentence "revealed that ASXL2 bound far fewer transcriptional start sites (TSS) than RUNX1 or AML1-ETO (Fig. 3d)." should be changed to the following or something similar, ""revealed that ASXL2 displayed less enrichment at transcriptional start sites (TSS) than RUNX1 or AML1-ETO (Fig. 3d).".
2. The motif analysis presented in should be altered in two ways. First, since the authors hand-picked motifs from this list the ranking of the motif picked should be added to the figure. Second, the PU.1 motif identified is a composite PU.1-IRF motif not solely the PU.1 motif the authors state in the label. This should be accurately displayed as a PU.1-IRF motif.

Reviewer #1

We thank the Reviewer for their careful review of the manuscript and for deeming the as "now acceptable for publication."

Reviewer #2

We thank the Reviewer for their careful review and for stating that we have "performed thorough revisions and addressed my concerns."

a. Major comment 6/Supplementary Figure 7c: The authors have examined the correlation between H3K27Me3 marks and differential gene expression upon ASXL2 knockdown. However, given the P values of 0.4 and 0.8, I am not sure they can state that there is a correlation.

We agree with the Reviewer and we have now removed the statement about correlations in gene expression and H3K27me3 loss from the manuscript. This section of this manuscript now states:

"There was no clear correlation between the loss of H3K27me3 upon ASXL2 knockdown and alterations in gene expression (Supplementary Figure 7c)."

b. Major comment 6/Figure 4f: It is difficult to visualize where on the HOXA gene locus the H3K27Ac peaks are altered. Could the authors perhaps highlight them by putting boxes around significantly altered peaks?

The *HOXA* locus in the SKNO-1 cells in Figure 4f is an example of a locus where the alterations in chromatin state are primarily evidenced by loss of H3K27me3 and increased H3K4me1. As explained in the prior response to review we did not attempt to give the impression that H3K27Ac is globally changed with ASXL2 loss or even that H3K27Ac alone is changed at enhancers. Given that the changes in H3K27me3 and H3K4me1 in Figure 4f with ASXL2 loss are evident without boxes, we have respectfully chosen not to add any further designations such as boxes onto Figure 4f.

c. Minor comment relating to Figure 2d/Supplementary Figure 4a: There still seems to be some confusion about the percentages displayed in the FACS plots in the bottom half of Supplementary Figure 4a ("Gated on LSK from the above"); the authors state that the figure shows the percentages of each population from the parent gate. In the bottom figures, the numbers are not the percentages of the parent "LSK" gate but of the grandparent "live, CD45/2+, lineage-negative cells" gate. Please could the authors amend this, as the CD150+ CD48- HSC fraction appears increased as a proportion of the LSK in the *Asxl2* and *Asxl1/Asxl2* knockout mice at the expense of the CD48+ CD150- fraction and this should be made clearer.

We thank the Reviewer for pointing this out and we have now amended the figure legend for Supplementary Figure 4a to note that all frequencies shown in Supplementary Figure 4a are percentages of live, CD45/2+, lineage-negative cells.

Reviewer #3

We thank the Reviewer for their careful review and for stating that the manuscript is "well revised and improved " and addresses all of their concerns.

1. In regards to the overlap of *Asxl2* signal at TSS presented in Figure 3d, the histogram presented should be interpreted as enrichment and not binding. In the text on pg 9 the sentence "revealed that ASXL2 bound far fewer transcriptional start sites (TSS) than RUNX1 or AML1-ETO (Fig. 3d)." should be changed to the following or something similar, ""revealed that ASXL2 displayed less enrichment at transcriptional start sites (TSS) than RUNX1 or AML1-ETO (Fig. 3d)."

We thank the Reviewer for this point and we have revised the sentence as the Reviewer rightfully

recommended.

2. The motif analysis presented in should be altered in two ways. First, since the authors hand-picked motifs from this list the ranking of the motif picked should be added to the figure.

Second, the PU.1 motif identified is a composite PU.1-IRF motif not solely the PU.1 motif the authors state in the label. This should be accurately displayed as a PU.1-IRF motif.

We agree with the Reviewer on this point and we have revised Supplementary Figure 5d to indicate the rank of each motif and the name of the composite PU.1-IRF motif.